# Design and realization of compressor data abnormality safety monitoring and inducement traceability expert system

Yuan Wang[1], Shaolin Hu [2]*

1 School of Information and Control Engineering Jilin Institute of Chemical Technology Jilin, Jilin, China,
2 Automation School Guangdong University of Petrochemical Technology, Maoming, Guangdong, China

* hfkth@126.com

## Abstract

Centrifugal compressors are widely used in the oil and natural gas industry for gas compression, reinjection, and transportation. Fault diagnosis and identification of centrifugal compressors are crucial. To promptly monitor abnormal changes in compressor data and trace the causes leading to these data anomalies, this paper proposes a security monitoring and root cause tracing method for compressor data anomalies. Additionally, it presents an intelligent system design method for fault tracing in compressors and localization of faults from different sources. This method starts from petrochemical big data and consists of three parts: fault dynamic knowledge graph construction, instrument data sliding fault-tolerant filtering, and the fusion and reasoning of fault dynamic knowledge graph and instrument data variation monitoring. The results show that this method effectively overcomes the problems of false alarms and missed alarms based on fixed threshold alarm methods, and achieves 100% classification of two types of faults: non starting of the drive machine and low oil pressure by constructing a PCA (Principal Component Analysis)—SPE (Square Prediction Error)—CNN (Convolutional Neural Network) classifier. Combined with dynamic knowledge graph and NLP (Natural Language Processing) inference, it achieves good diagnostic results.

## 1. Introduction

In actual production activities, the occurrence of system failures is inevitable due to internal factors (such as part damage) or environmental factors (such as ventilation and sealing). The compressor system consists of a lubrication system, cooling system, power system, and other components. These complex systems, composed of many interconnected units, can significantly impact normal operations when a failure occurs and may further lead to considerable losses that are difficult to estimate. Therefore, it is important to study the fault diagnosis methods of compressor systems. Expert systems usually consist of an inference engine, a knowledge base, an interpretation facility, and a user interface. In practice, most of the information required to create rules comes from expert experience or maintenance manuals. However, due

**Data Availability Statement:** All relevant data are within the paper and its Supporting Information files.

**Funding:** This work was supported by the Natural Science Foundation of China (62373115); Nature and Science Foundation of Guangdong Province

(2023A1515012341,2024A1515010870). The funders had no role in study design, data collection and analysis, decision to publish, or preparation of the manuscript.

**Competing interests:** The authors have declared that no competing interests exist.

to the lack of comprehensive experience and information, this presents some inherent constraints on the development of the rule base. Limited knowledge acquisition is a bottleneck for traditional rule-based expert systems, restricting their scope of application. Additionally, traditional rule-based fault diagnosis expert systems lack flexibility.

In the petrochemical production process, instrumentation is not only an important basis for understanding the state of equipment, operating conditions, material quality and emissions, but also an important basis for monitoring the operation of the entire device faults and potential accidents, and discovering abnormal working conditions and disaster risks. In recent years, data-driven methods based on deep learning [1–6] have made significant progress in fault diagnosis of complex systems due to their ability to efficiently and economically collect large amounts of data. For example, the method of combining Z numbers with fuzzy logic improves the accuracy results of expert systems applying risk analysis [7]. Combined with intuitionistic fuzzy set theory and expert-inspired frameworks, TFTs of dynamic systems with uncertain data can be quantitatively analyzed [8]. A methodology for constructing a knowledge base for an operational guidance expert system based on HAZOP analysis and accident analysis [9]. Artificial neural network based fault diagnosis system for detecting combustion faults, wear faults, fuel injection faults [10]. Analyzing the root cause of faults using the degree of change and the average variable threshold limit [11]. Combining multi-scale principal component analysis with signature based directed graph method for process monitoring and fault diagnosis [12]. Application of Threshold Recursive Graph and Texture Analysis in Diagnosing Static Friction in Process Control Circuits [13]. A Multi Kernel Support Vector Machine (MK-SVM) algorithm is proposed for diagnosing simultaneous faults in distillation towers [14]. Combining multivariate index weighted moving average (MEWMA) monitoring scheme with PCA modeling to improve anomaly detection performance [15]. Single class support vector machine (KPCA-OCSVM) based on kernel principal component analysis and various kernels to learn an anomaly free training set, and then classify the test set [16]. An engineering based approach that can be improved upon traditional natural language processing to construct a domain knowledge graph based on the ontology of petroleum exploration and development [17]. Addressing the issues of label dependency, sparse interaction data, and irrelevant knowledge in community-based software expert recommendations, paper propose an Expertise Preference-Aware Network model for Software Expert Recommendation (EPAN-SERec) with knowledge graph [18]. Adopting knowledge graph best development practices to achieve scalable and optimized manufacturing processes [19] etc. However, because there are many types of equipment instruments in the petrochemical process industry, there are many hidden dangers that affect compressor safety, and there are many signs and manifestations of compressor failure, there are mostly complex interlocking relationships, feedback relationships, or causal relationships between instrument data and failures. The effects of faults are intertwined and change dynamically over time, making it difficult for conventional fault tree analysis, cause-and-effect tree analysis and simple hierarchical network topology diagrams to fully reflect their complex multiple couplings and mutual effects, which increases the complexity of the compressor and Difficulty of process safety risk analysis.

In the production process of the existing large-scale petrochemical plant, whether the monitoring platform adopts the DCS mode or the FCS mode, due to the above-mentioned complexity and technical difficulty, the overrun alarm based on the fixed threshold or the overrun grading alarm method based on the multi-level threshold that is most widely used in the field of instrument data anomaly monitoring is not only difficult to set accurately, but also the monitoring method itself is difficult to avoid the limitations such as false positives, false alarms or missed judgments and missed alarms. Especially in the early stage of the occurrence of the abnormality or when the amplitude of the abnormality is less than 3 times the variance of the

measurement error, it is difficult to detect the early fault signs and the weak fault is difficult to detect.

To this end, this paper proposes a compressor data security monitoring and trigger source method based on a dynamic knowledge graph to break through the weak links in existing petrochemical instrument safety monitoring technology and traditional expert systems, and solve the problem of difficulty in separating instrument abnormalities from equipment failures. It is difficult to catch small faults in time and the problem of frequent false alarms and missed alarms in the monitoring process is of great significance to ensuring petrochemical safety.

In summary, the main contributions of this study are as follows:

Based on the sample data obtained by sliding window processing, a sliding window three-segment mid-section fault-tolerant filtering method is implemented to implement a safe and reliable sliding window fault-tolerant filter, as well as a residual generation and data mutation monitoring algorithm based on fault-tolerant filtering.(2)Based on the fault data obtained by sliding window processing, a deep learning model for system fault classification combining PCA-SPE and CNN is proposed to realize 100% classification of compressor faults.(3)Combination of model-based classification results and expert system realizes dynamic knowledge graph-based instrumentation data security monitoring and causative origins.

## 2. Construction of a residual generation and data variance monitoring algorithm based on fault-tolerant filtering

Accurate extraction of intrinsic variations in instrument data is an important technical aspect of accurate monitoring of data variations and early detection of small variations. For the petrochemical instrument sampling data sequence $\{Y(t_i), i = 1,2,\ldots,N\}$, this paper proposes and implements a safe and reliable sliding window fault-tolerant filter, as well as a residual generation and data mutation monitoring algorithm based on fault-tolerant filtering:

Design a sliding window three-segment mid-section fault-tolerant filtering method: Specifically, construct a sliding window with window width $s = 3k+2$, where $k$ is an adjustable parameter (the default value is 5), For the instrument sampling data fragments in the half-open interval time window $[t_i, t_{i+s})$, $X_i = \{Y(t_j), t_j \in D_i\}$ are sorted in order from small to large, The sorted data fragment is $\tilde{X}_i = \{\tilde{Y}(t_j), t_j \in D_i\}$, and the mean filter $Y_i^{med}$ and filter residuals $E_i^{med}$ of $k$ points in the middle of $\tilde{X}_i$ are calculated.

$$Y_i^{med} = \frac{1}{k}\sum_{j=1}^{k}\tilde{Y}(t_{i+k+j}), \; E_i^{med} = \frac{1}{k}\sum_{j=1}^{k}(\{Y((t_{i+k+j}) - Y_i^{med}\}) \tag{1}$$

The above sliding filter is strongly tolerant to abnormal data Outliers and localized spots. Briefly note that the residual sequence formed by the residuals obtained by the above sliding window fault-tolerant filter is $\{E_i^{med}, i = 1, 2, \ldots\}$, and the threshold parameter $C_i$ is constructed by using a recursive algorithm

$$C_i = 3\sqrt{\frac{2}{i(i+1)}\sum_{j=1}^{i}j\phi_j(E_j^{med}, C_{j-1})^2} \tag{2}$$

where $\frac{2}{i(i+1)}$ indicates that two permutations are arbitrarily selected at $i+1$ time points, and their weights decrease with the increase of $i$ when the statistics of multiple points are accumulated. $j$ is the weight coefficient, which means that the later the moment $j$ has a greater effect on the threshold value. $\varphi_j(E_j^{med}, C_{j-1})$ is a piecewise function used to adjust the relationship between $E_j^{med}$ and $C_{j-1}$, as shown in Formula (3), $E_j^{med}$ represents the filtered residual result of the low j-

th calculation in the residual sequence, and $C_{j-1}$ represents the threshold value of the j-1st calculation.

$$\phi_j\left(E_j^{med}, C_{j-1}\right) = \begin{cases} E_j^{med}, & |E_j^{med}| \leq 3C_{j-1} \\ -3C_{j-1}, & E_j^{med} < -3C_{j-1} \\ 3C_{j-1}, & E_j^{med} > 3C_{j-1} \end{cases} \tag{3}$$

And based on this, monitor and diagnose instrument data changes:

When $E_i^{med} \in [-C_i, C_i]$, the data is judged to be normal; When $E_i^{med} > C_i$, the data is suspected to be abnormal, when $E_i^{med} > 1.5C_i$, data reports a high-level abnormality, and when $E_i^{med} > 3C_i$, data reports a very high-level abnormality. When $E_i^{med} < -C_i$, data is suspected to be abnormal; when $E_i^{med} < -1.5C_i$, data reports a low-level abnormality; when $E_i^{med} < -3C_i$, data reports a very low-level abnormality. The complete implementation process is shown in Fig 1.

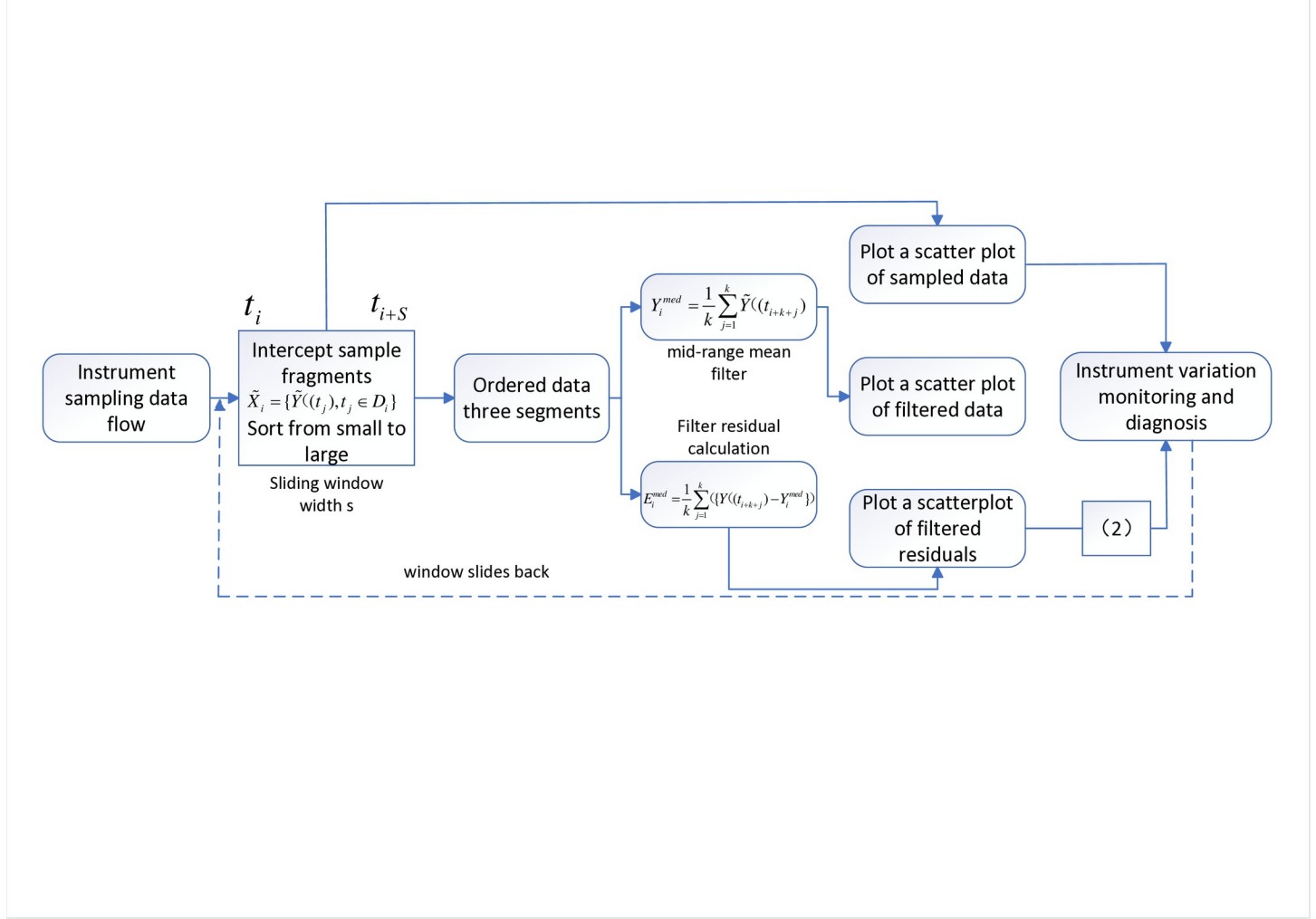

**Fig 1. Schematic diagram of instrument variation monitoring process.**

## 3. Construction of a deep learning based system fault classifier

The framework of the system fault classifier based on deep learning proposed in this paper is composed of two parts: PCA-SPE monitoring and CNN classifier. Due to the significant differences in the data selected in this article, it is difficult to distinguish between known edge normal data and abnormal data. Therefore, first, PCA-SPE is used to screen the data for fault samples and obtain corresponding fault samples. Then, combined with the constructed CNN classifier, the fault classification is achieved. The detailed construction process of the corresponding algorithm is as follows.

### 3.1 PCA-SPE-based fault monitoring

Firstly, according to the offline data, the data $H$ is normal, $H$ is $H(t_1)...H(t_{1500})$, and $H$ is 12, and the dimension is the same as that of the monitoring sample. Then the matrix $H$ is subjected to error tolerant filtering [20], and the normalization process is carried out, and the data of each column of matrix $H$ are normalized to the data with a mean value of 0 and a root mean square of 1 to obtain the matrix $\hat{X}$.

The samples used for monitoring are obtained continuously from $\{Y(t_i)i = 1,2,...,N\}$ through a sliding window. $Y(t_i)$ is 12-dimensional data. The data segment with a sliding window of 100 is recorded as $Y[t_k:t_{k+100}]$. The data $Y[t_k:t_k+100]$ is normalized, the mean, and variance are the mean and variance of the training data, and each column of data in the matrix $Y$ is normalized to the data with a mean value of 0 and a root mean square of 1 to obtain matrix $\hat{Y}$.

1) Calculate the covariance matrix of the normalized training data and the monitoring data respectively:

$$R_a = \frac{1}{n-1}\hat{X}^T\hat{X} \tag{4}$$

$$R_s = \frac{1}{g-1}\hat{Y}^T\hat{Y} \tag{5}$$

where n is the number of the training samples, $g$ is the number of the monitoring samples, and the resulting covariance matrix is a characteristic 12*12-dimensional matrix.

2) The eigenvalues are sorted from large to small, and the top k features with a sum of more than 85% eigenvalues are selected for PCA dimensionality reduction.

$$\frac{\sum_{i=1}^{k}\lambda_i}{\sum_{i=1}^{m}\lambda_i} \geq 0.85 \tag{6}$$

The first k eigenvalues from large to small form a diagonal matrix $S_{k*k}$, and k corresponding eigenvectors form a matrix $P_{m*k}$.

The training set diagonal matrix and the corresponding eigenvectors form the matrix:

$$S_{k_1*k_1} = diag(\lambda_1, \lambda_2, \cdots, \lambda_{k_1}) \tag{7}$$

$$P_{m*k_1} = [p_1, p_2, \cdots, p_{k_1}] \tag{8}$$

The diagonal matrix of the monitoring set and the corresponding feature vector form a matrix:

$$\hat{S}_{k_2*k_2} = diag(\hat{\lambda}_1, \hat{\lambda}_2, \cdots, \hat{\lambda}_{k_2}) \tag{9}$$

$$\hat{P}_{m*k_2} = [\hat{p}_1, \hat{p}_2, \cdots, \hat{p}_{k_2}] \tag{10}$$

For $\hat{X}$, $\hat{Y}$ after dimensionality reduction, the number of samples remains unchanged, but the number of features becomes K. After dimensionality reduction, it is:

$$\breve{X}_{n*k1} = \hat{X}_{n*m} * P_{m*k_1} \tag{11}$$

$$\breve{Y}_{g*k_2} = \hat{Y}_{g*m} * \hat{P}_{m*k_2} \tag{12}$$

The formula for calculating the matrix of $X^\frown$, $Y^\frown$ obtained by reconstructing $\hat{X}$ and $\hat{Y}$ is as follows:

$$\widehat{X} = \breve{x}_{n*k_1} P_{m*k_1}^T = \hat{X}_{n*m} P_{m*k_1} P_{m*k_1}^T \tag{13}$$

$$\widehat{Y} = \breve{Y}_{g*k_2} \hat{P}_{m*k_2}^T = \hat{Y}_{g*m} \hat{P}_{m*k_2} \hat{P}_{m*k_2}^T \tag{14}$$

where n is the number of training samples, $g$ is the number of monitoring samples, m is the number of corresponding key variables, and k is the number of corresponding variables after dimensionality reduction.

The training sample used in this paper is 1500 sampled values and the monitoring sample window is a large sample of 100 data and the two samples are independent of each other and conform to a normal distribution and $\hat{X}$ conforms to a normal distribution with zero mean. We used a PCA-based process monitoring methodology with the squared prediction error (SPE) statistic [21, 22] to make the determination.

The control limits for the SPE statistic were proposed by Jackson and Mud Holkar in 1979. $\widehat{Y}$ conforms to the multivariate normal distribution, and the SPE statistic [23] is established for $\widehat{Y}$, that is, for $\widehat{Y}_{i,:}$ (i = 1,2... g) Construct statistic Q:

$$Q = \widehat{Y}_{i,:} * (I_{m*m} - \hat{P}_{m*k_2} \hat{P}_{m*k_2}^T) * \widehat{Y}_{i,:}^T \tag{15}$$

where $\widehat{Y}_{i,:}$ is the row vector of row i of $\widehat{Y}$, $I_{m*m}$ is the identity matrix, and $\hat{P}_{m*k_2}$ is the matrix $\hat{P}_{m*k_2}$ of k eigenvectors corresponding to k eigenvalues from large to small from (1) and (2).

Since $\widehat{X}$, $\widehat{Y}$ obeys a normal distribution with zero mean, and the sample sampling points are independent of each other, it has been proved in the literature [24] that Eq (13) is approximately in line with the normal distribution.

$$(Q/\theta_1)^{h_0} \sim N\left[1 + \frac{\theta_2 h_0 (h_0 - 1)}{\theta_1^2}, \frac{2\theta_2 h_0^2}{\theta_1^2}\right] \tag{16}$$

$$其中 \theta_r = \sum_{j=k1+1}^{m} \lambda_j^r, r = 1, 2, 3 \tag{17}$$

$$h_0 = 1 - \frac{2\theta_1 \theta_3}{3\theta_2^2} \tag{18}$$

Therefore, the approximate mean of $c$ is a normal distribution of 0, which is calculated as

follows:

$$c = \frac{\theta_1[(Q/\theta_1)^{h_0} - 1 - \theta_2 h_0(h_0 - 1)/\theta_1^2]}{\sqrt{2\theta_2 h_0^2}} \tag{19}$$

Then the control limit of Q is:

$$Q_\alpha = \theta_1 \left[\frac{c_\alpha h_0 \sqrt{2\theta_2}}{\theta_1} + 1 + \frac{\theta_2 h_0(h_0 - 1)}{\theta_1^2}\right]^{\frac{1}{h_0}} \tag{20}$$

$c_\alpha$ is the confidence limit of the standard normal distribution, $c_\alpha$ is 0.996 confidence, $\lambda$ is the eigenvalue of the corresponding covariance matrix $R_{ta}$, and $h_0$ is the adjustment coefficient.

 1) According to Eqs (15) and (20), the SPE statistic $Q$ and the statistical limit $Q_\alpha$.

 2) Fault determination

 If the system is functioning normally, the $Q$-value of the sample should meet $Q < Q_\alpha$, otherwise it is considered to be faulty.

 If $Q < Q_\alpha$, the corresponding fault data $\tilde{Y}$.

## 3.2 CNN module

The CNN module consists of a CNN layer, a pooling layer, and a fully connected layer, as shown in Fig 2, which are described in detail below.

 The CNN layer consists of two operations, the convolution operation and the activation operation. The input to the convolution operation is a 3D data $Z \in R^{w \times h \times c}$, where w, h, and c represent its width, height, and number of channels, respectively. In particular, when c = 1, it is reduced to 2D data as input to the proposed model $\tilde{Y} \in R^{w \times h \times c}$. Specifically, the horizontal

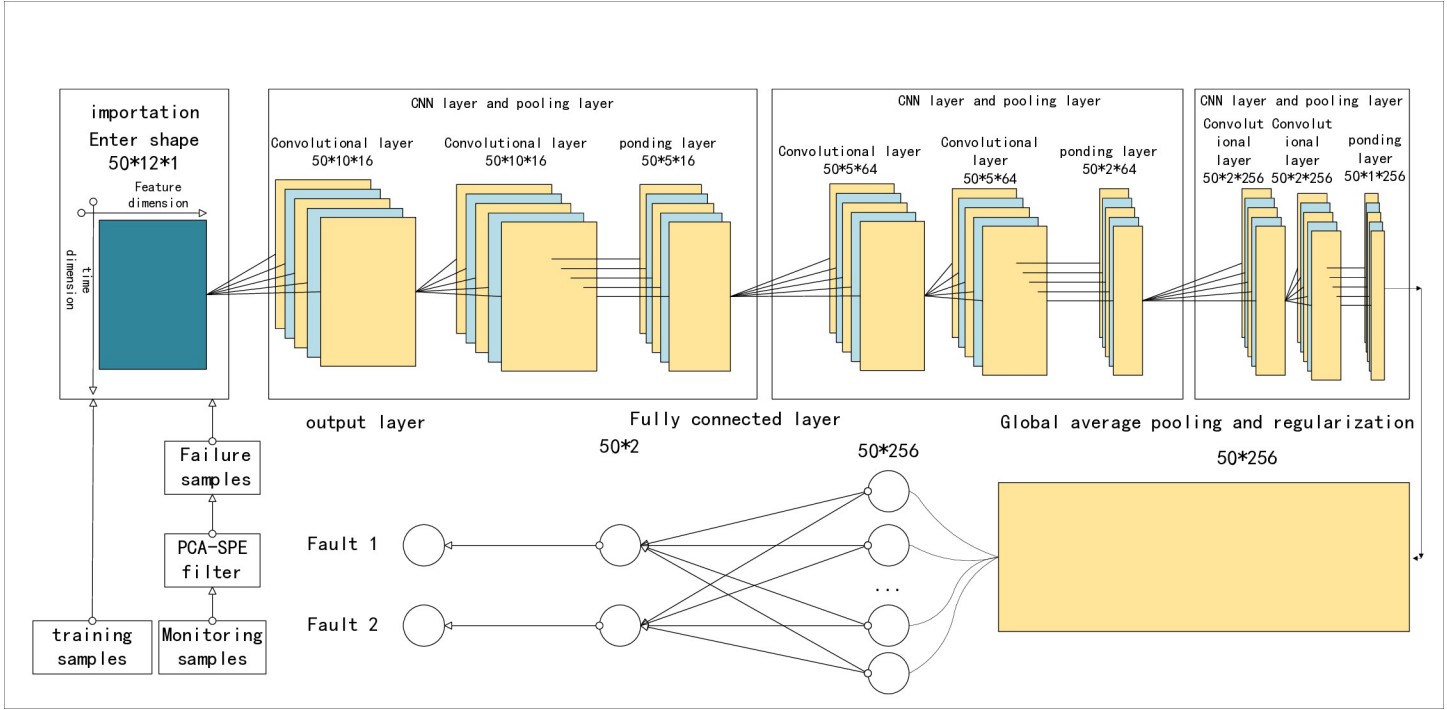

**Fig 2. Structure of PCA-SPE-CNN classifier.**

axis of two-dimensional data represents the feature dimension, and the vertical axis represents the time dimension. The axis scale is characterized by the number of attributes and the width of the sliding window, respectively. In order to clearly illustrate the convolution operation, we first define four hyperparameters, namely the number of convolution kernels K, the size of each convolution kernel F, the step size of the convolution S and the number of zeros P. Then, for the convolution operation of each convolution kernel, the data $\tilde{Y} \in R^{w \times h \times c}$ is filled with zeros to ensure that the shape of the output and input are the same, resulting in the transformed data $\tilde{Y}^* \in R^{(w+p) \times (h+p) \times c}$. After that, the slicing operation is:

$$\tilde{Y}_{t_w, t_h} = \tilde{Y}^*[t_w : t_w + F, t_h : t_h + F, :] \tag{21}$$

where $t_w : t_w + F$ represents the operation of extracting the subset located between row $t_w$ and row $t_w + F$, $t_h : t_h + F$ represents the operation of extracting the subset located between column $t_h$ and column $t_h + F$, and : represents all the data in the direction of the cutting channel. $t_w$ and $t_h$ start at 1 and change with the number of steps S. According to $(t_w, t_h)$, data $\tilde{Y}_{t_w, t_h}$ can be extracted from $\tilde{Y}^*$. Let the convolution kernel be denoted as kernel$\in R^{F \times c}$, then $T_{t_w, t_h} \in R^{F \times c}$ is defined as kernel and $\tilde{Y}_{t_w, t_h}$ multiplied element by element, and the formula is as follows:

$$T_{t_h} = Kernel \times \tilde{Y}_{t_h} \tag{22}$$

Therefore, $b_{t_h}$ can be obtained through the following methods:

$$b_{t_h} = \sum_{j=0}^{F} \sum_{k=0}^{C} T_{t_w, t_h}[j, k] \tag{23}$$

where $T_{t_h}[j, k]$ denotes the element of $T_{t_w, t_h}$, and $\tilde{B} \in R^{w* \times h*}$ can be obtained from $b_{t_w, t_h}$ ordered by the size of $t_w, t_h$. It can be noted that

$$h^* = \frac{h - F + 2*P}{S} + 1 \tag{24}$$

For each convolutional kernel, a channel such as B is obtained, and then the above operation is repeated on K convolutional kernels to get a new 3D data $S \in R^{w* \times h* \times K}$. After the convolution operation, the activation operation is essential. It enables the network to obtain nonlinear expressions of the input to enhance representation capabilities and make the learned features more separable. Use the Rectifier Linear Element, which is a widely used linear element, which can activate each element of the $S$ in the following ways:

$$A[i, j, k] = f(S[i, j, k]) = \max\{0, S[i, j, k]\} \tag{25}$$

where $S[i,j,k]$ is the element of S and $A[i,j,k]$ is the element of $A \in R^{w* \times h* \times K}$, the data obtained after the activation operation.

Dropout regularization:

The Dropout layer reduces model overfitting by randomly zeroing out a portion of the elements of the input data. The formula for the operation of Dropout is as follows:

$$\text{Dropout}(Z, p) = Z \cdot M \tag{26}$$

where Z is the input data, p is the discard rate, and M is the mask matrix generated from the Bernoulli distribution.

Fully Connected Layer (Dense):

The fully connected layer maps the output of the globally averaged pooling layer to the probability of the output class through a weight matrix and bias vectors. The operation formula for the fully connected layer is as follows:

$$\text{Dense}(\hat{Z}, W, b) = \text{softmax}(\hat{Z} \cdot W + b) \tag{27}$$

where $\hat{Z}$ is the output of the global average pooling layer, W is the weight matrix, b is the bias vector, and softmax $(\cdot)$ is the softmax function to normalize the raw scores into category probabilities.

## 4. Fault diagnosis based on expert system

In the actual situation of compressor online monitoring, because the parameters are outside the allowable setting range, a large amount of information has surfaced before the alarm, which is identified by PCA-SPE-CNN in this paper. This study explores the correlation of deviating parameters, equipment components, and failure modes based on the data collected in this paper, which is derived from published literature and compressor data from PetroChina West-East Gas Pipeline Co., Ltd. at Gaoling Station. Regarding the diagnosis and traceability issues of compressors, a network-based intelligent diagnostic program was developed to find abnormalities by processing monitoring data. Once the early failure mode is identified, the program will use the knowledge base to try to infer the possible failure and achieve fault traceability and location.

### 4.1 Construction of knowledge graph

Constructing a dynamic knowledge graph is the premise and foundation for identifying the causes of instrument variation. This paper proposes a bottom-up method for constructing a dynamic knowledge graph of petrochemical instrument data variations: based on petrochemical big data, a petrochemical data flow structure classifier is established to extract structured data, semi-structured data and unstructured data from petrochemical big data. structured data; Perform preprocessing such as cleaning, deduplication, and formatting on the data stream to form corresponding expandable data sets of structured, semi-structured, and unstructured data; Extract knowledge from the data set to form an entity set E of materials, working conditions, modalities, and production processes related to petrochemical instruments, a relationship set R that reflects process relationships, structural relationships, and association relationships, as well as a relationship set R that reflects the quantification degree of process conditions. An attribute set A such as the object's normality and so on; Using the three sets of E, R, and A, the triplet {(ě,§,ī): ě,ī∈E,§∈R} and {(ě, ζ, ī): ě,ī∈E; ζ ∈A}, and the entity disambiguation, co-signifier dissolution and knowledge merging are carried out on the above-mentioned triplet sets to achieve knowledge fusion; Define and construct the concepts, terms and relationships in the petrochemical instrument fault knowledge map to form a fault knowledge base structure for petrochemical instruments; Expand existing knowledge about instrument faults and obtain new knowledge through information extraction, knowledge fusion and knowledge reasoning; A graph database is used to store the obtained knowledge graph; While continuously receiving the instrument data and process data sent by the petrochemical DCS system, combined with the incremental data information, the iterative optimization of the knowledge graph is realized, the iterative optimization in this article is mainly achieved by constructing a PCA-SPE-CNN classifier to classify faults in real-time monitored instrument data, thereby realizing inference iteration, and combining expert knowledge in the knowledge graph to update and iterate knowledge.The dynamic knowledge graph of the in-service process

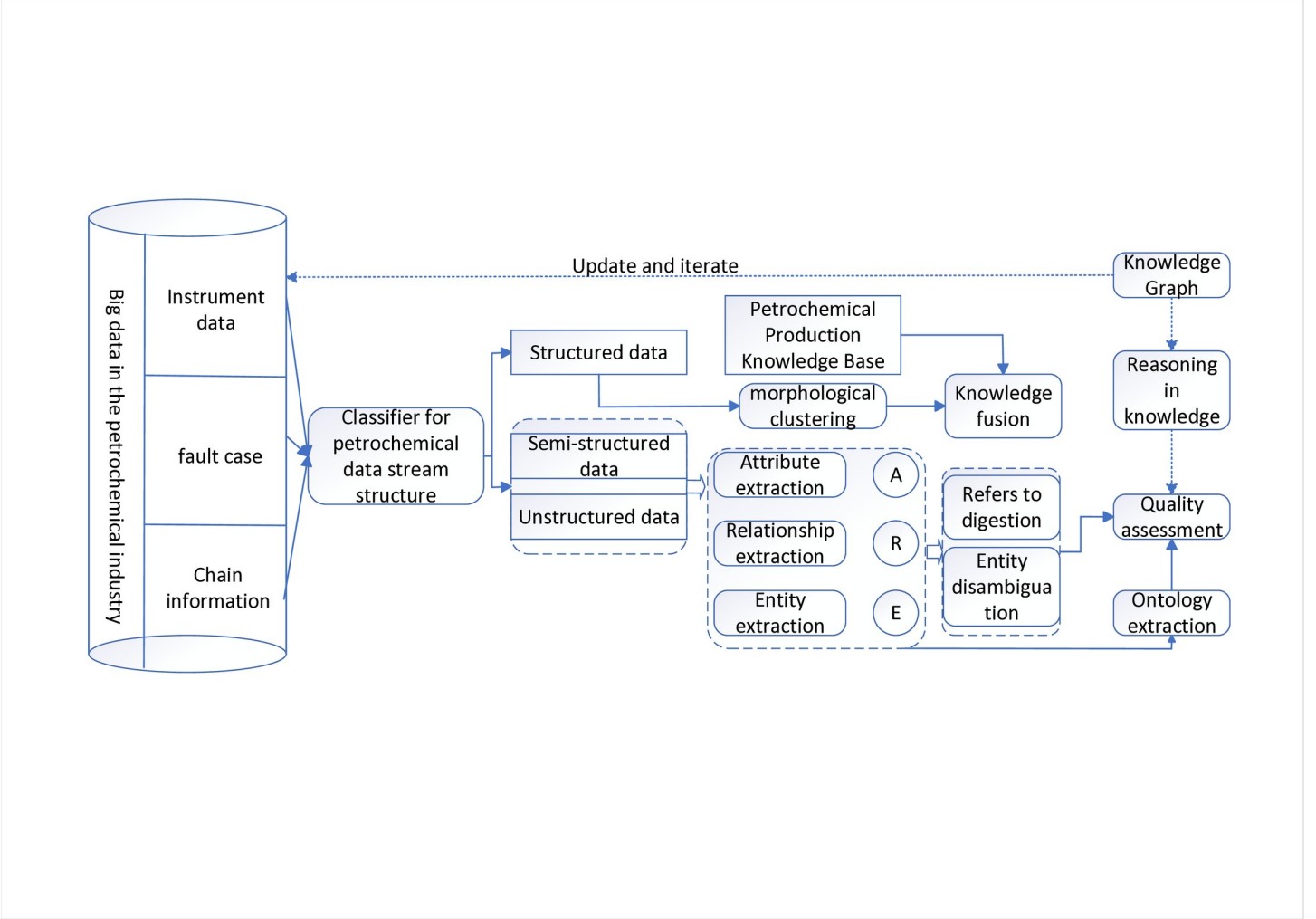

**Fig 3. Schematic diagram of the construction process of petrochemical instrument knowledge graph.**

of the petrochemical instrument is completed, the technical process is shown in Fig 3, and the construction schematic diagram is shown in Figs 4 and 5.

## 4.2 Construction of a compressor failure cause traceability system

Identifying the source of variations in instrument data is critical to preventing fail-safe risks. This paper proposes a multi-induced anomaly traceability method for petrochemical instrument data based on improved CNN classifier, deep integration of natural language processing and dynamic knowledge graph, which uses dynamic knowledge graph and integrates intelligent information processing technologies such as cognitive computing, feature extraction, knowledge representation, knowledge reasoning, and information retrieval, as well as convolutional neural network and natural language processing model to realize the source of inducement of instrument data variation. Specifically, the method of inducing data mutation sources consists of the following three parts:

1) Inference engine design based on natural language processing NLP. Based on the above dynamic knowledge graph, a collection of heterogeneous knowledge graph triples describing

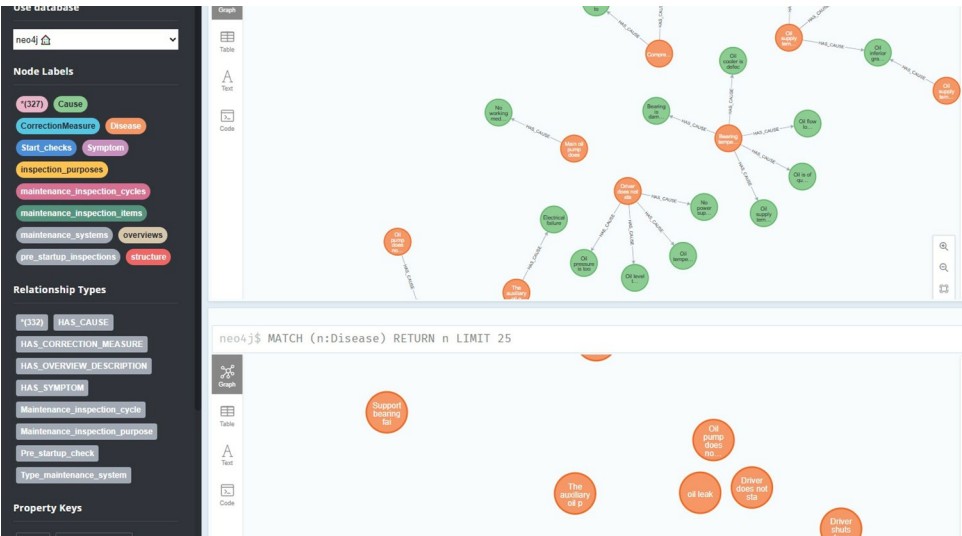

**Fig 4. Schematic diagram of node relationships in the petrochemical instrument knowledge graph.**

the fault sources of petrochemical equipment and compressors, the heterogeneous patterns of instrumentation data, and the relationship between the fault sources and the data heterogeneity are constructed to form the graph dataset; A graph dataset is used as the basis for modeling instrumentation data variability and compressor faults in combination with an improved CNN classifier, and a priori logic rules are introduced in the modeling process; Based on the historical data of petrochemical instruments and the dynamic knowledge graph of faults, the summary module of the relationship between given entities and the inference module of state inference based on this are the inference modules. Relational NLP is constructed as an editor to realize the specific classification learning of the relationship between faulty entities and neighboring entities in the knowledge graph, and combined with the set question words, the matching is more efficient to obtain the corresponding classification representation, and then the NLP inference engine based on dynamic knowledge graph is constructed.

2) Inference engine design for multi-mode fusion. In this paper, an inference method based on multi-mode fusion of inference layer is designed to realize the source and location of

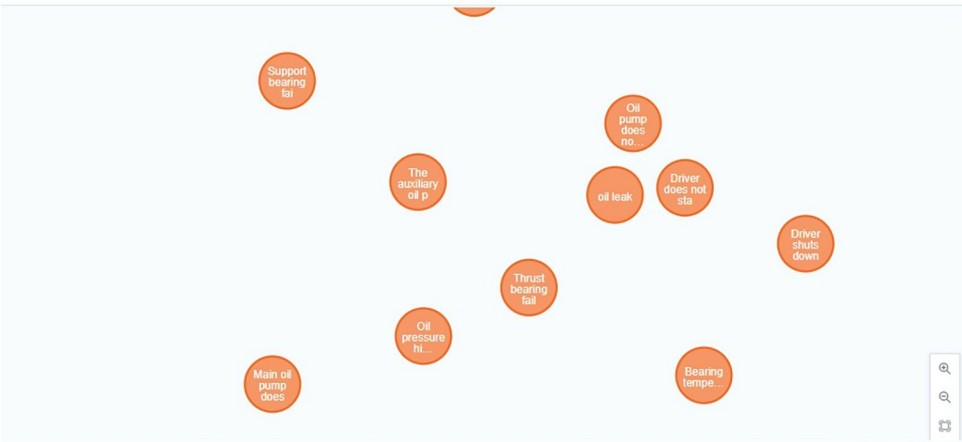

**Fig 5. Node entity diagram of petrochemical instrument knowledge graph.**

instrument and equipment faults. Specifically, a dynamic knowledge graph-based NLP inference engine is used to construct a premise set and a consequence set for failure risk; The historical data and historical fault cases of instruments and equipment are used for model training to obtain the fuzzy correlation of the linkage between the premise set and the consequence set; Based on the equipment data, a supervised machine learning method is used to establish a fault detection and diagnosis model based on resolved redundancy, from which fault types are determined; Absorbing the technical kernel of fault inference engine based on NLP computing, we establish a natural language inference engine based on combining semantic relevance and premise set for fault tracing, and integrate it with machine learning model inference engine and NLP computing inference engine at the inference layer to realize fault tracing and localization. The above reasoning layer fusion diagnostic reasoning method can effectively avoid the limitations of conventional forward reasoning and reverse reasoning which are difficult to be used in situations such as causal inter-network correlation, and effectively overcome the influence of multiple coupling mutual cause and effect relationships between detection devices as well as between detection devices and the inspected object and the controlled process, and realize the traceability and localization of the instrumentation data and the equipment faults.

3) Design of multi-induced anomaly traceability system based on the fusion of instrument data and dynamic knowledge graph. Starting from petrochemical big data, through the construction of dynamic knowledge graph of instrument faults, the sliding fault-tolerant filtering of instrument data, and the integration and reasoning of dynamic knowledge graph of instrument faults and instrument data anomaly monitoring, a safety monitoring and inducement traceability system for instrument data anomalies is established, and an intelligent support platform is formed to serve the source of instrument safety detection and inducements.

## 5. Analysis of experimental results

This section mainly uses the data provided by China Petroleum West East Gas Pipeline Co., Ltd. for the Gaoling Station compressor to analyze the experimental data of motor speed, motor power, motor stator temperature, and lubricating oil main pressure. Using the above data for the analysis and validation of the algorithm, the detailed settings and analysis validation of the corresponding algorithm dataset are described below. The residual generation and data anomaly monitoring algorithm based on fault-tolerant filtering is mainly used for detecting the stationarity of each dimensional feature.The construction of classifier algorithms mainly considers the influence of twelve dimensional features on second-class faults.The construction of knowledge base based on knowledge graph is mainly used for fault tracing, and the specific analysis is as follows.

### 5.1 Residual generation based on fault-tolerant filtering and monitoring process analysis of data anomaly detection algorithm

Taking the structured data collected by the instrument, as well as the feature change data extracted from the semi-structured data and the unstructured data as the object, the sliding window fault tolerance rate wave algorithm (2) and the filter residual generation algorithm (3) established by the invention are used to realize the data-driven monitoring and diagnosis of the abnormal change of each feature state. He test dataset consisted of 428 samples. Each sample has 12 characteristics, the test sample points are 428*12 = 5136, and the data sampling frequency is 1 time/min. The schematic diagram of data sampling, filtering, and filtering residuals in the process of data anomaly monitoring is shown in Figs 6–8. The monitoring of each feature of the data is shown in Figs 9–11.

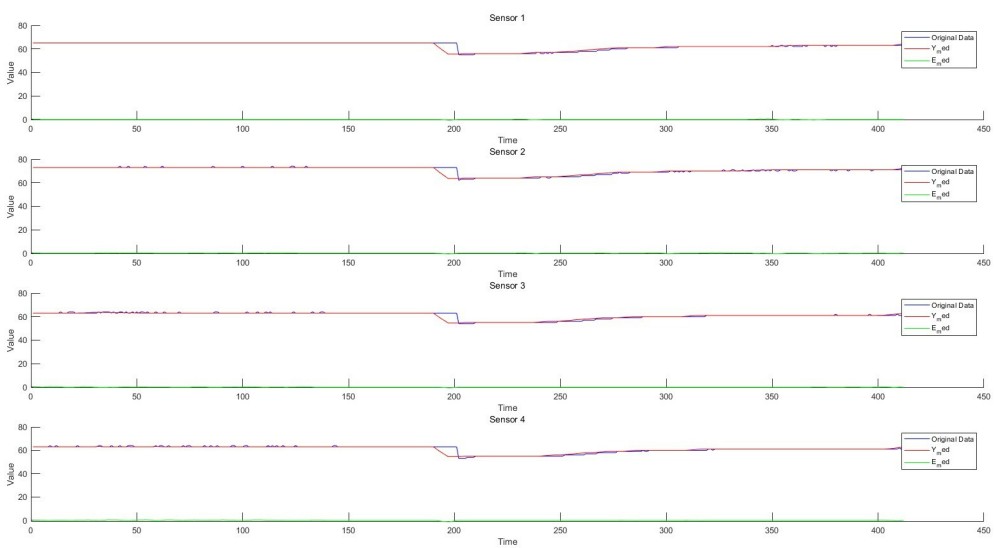

**Fig 6. Schematic diagram of residual of 1-4D filter.**

## 5.2 Introduction to classifier data settings

For each fault and normal state, the compressor status is reflected in a training data set and a test data set. Each training data set contains 1500 samples, and each test data set contains 228 samples. Each sample has 12 features, the original training sample points are $1500 \times 12 = 18000$, and the original test sample points are $228 \times 12 = 2736$. The data sampling frequency is 1 time/min. For the training samples containing faulty samples and normal samples, the samples are randomly sorted and normalized in order to ensure good learning results.

## 5.3 Analysis of data reasons

The motor fault and its related indicators are selected for verification. After the motor fault, the faulty motor cannot maintain a relatively constant speed and power as before the fault

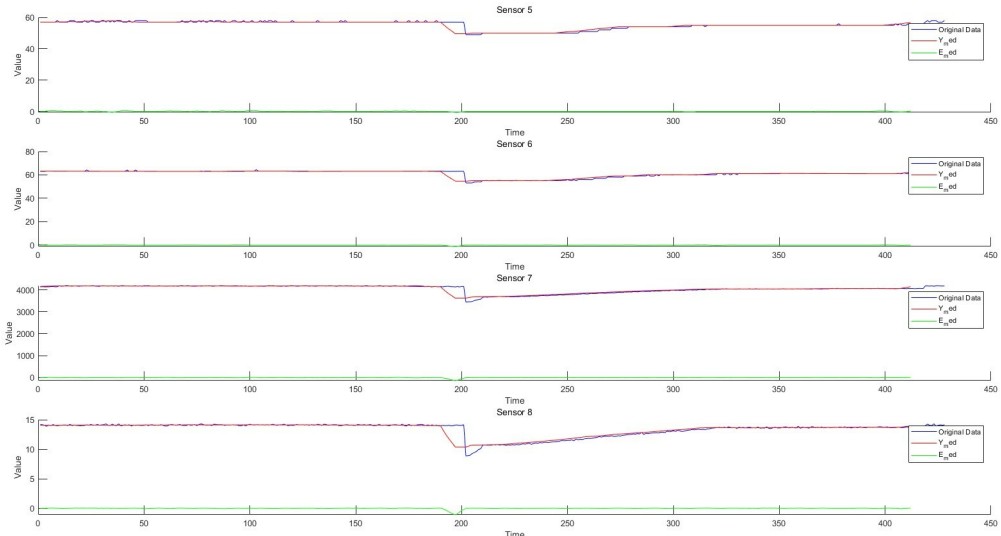

**Fig 7. Schematic diagram of residual of 5-8D filter.**

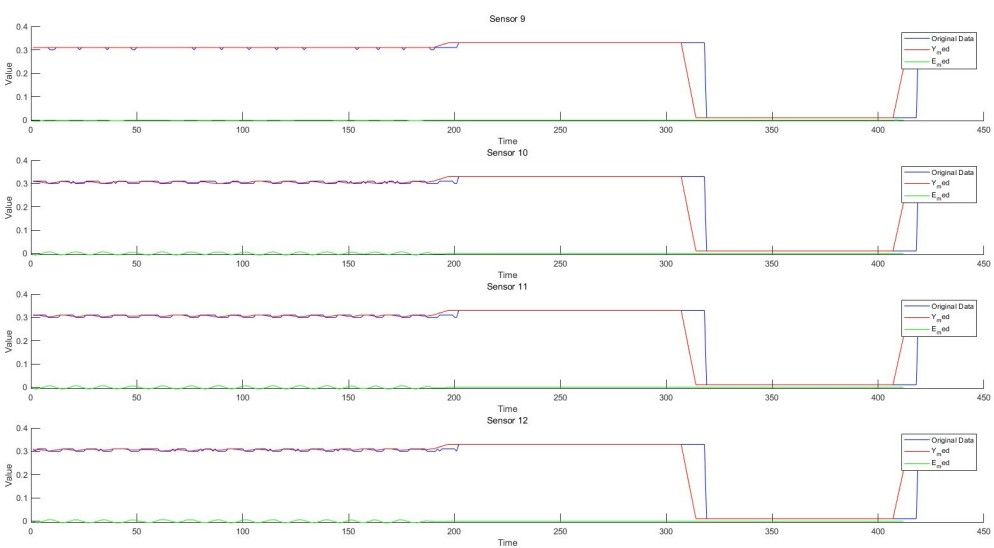

**Fig 8. Schematic diagram of residual of 9-12D filter.**

occurred, and the motor stator temperature also gradually changes with the motor fault. They can serve as key indicators of motor failure. The motor was in a shutdown state before 10:00 on 2022/10/14, and ran normally after 10:50. An abnormality occurred around 19:30, and returned to normal around 5:30 on 10/15. The temperature of the motor stator remains unchanged when the motor is stopped, rises after the motor operates normally, drops after an abnormality occurs in the motor, and recovers after the motor recovers. The motor speed, power diagram and motor stator temperature are shown in Figs 12 and 13. After the oil pressure failure, the lubricating oil main pipe pressure remained normal before 2:54 on 2022/10/23. An abnormality occurred after 2:54, and the main pipe pressure changed as follows As shown in Fig 14.

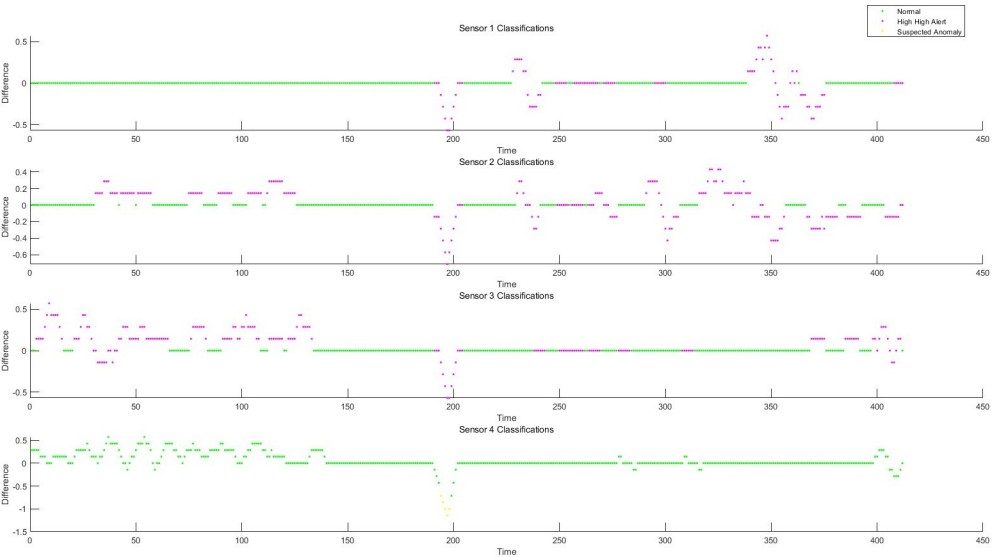

**Fig 9. Schematic diagram of 1–4 dimensional abnormal process monitoring.**

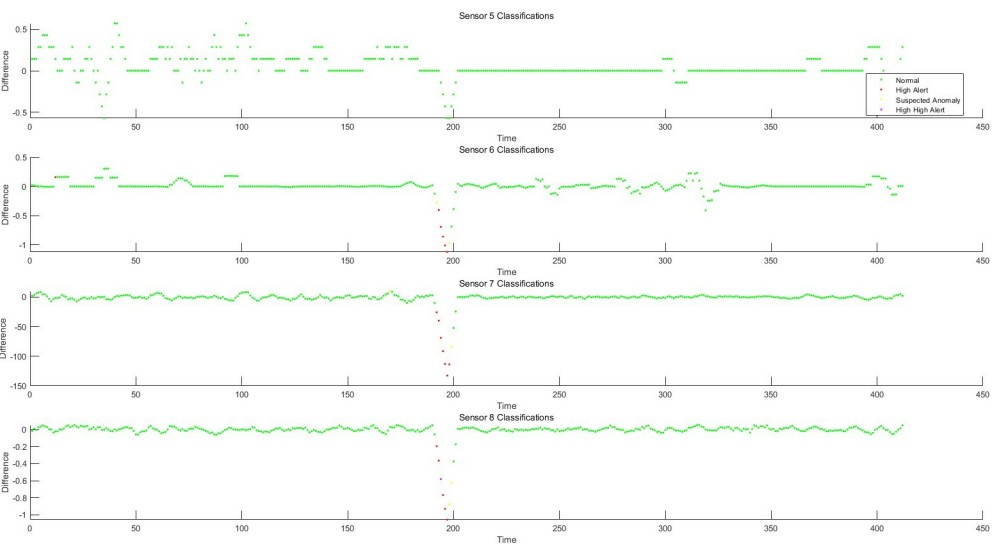

**Fig 10. Schematic diagram of 5–8 dimensional abnormal process monitoring.**

## 5.4 Network architecture setup

The input shape of the CNN network is batch 50 on the vertical axis, the matrix is feature 12 on the horizontal axis, the convolution kernel is 1*3, the number of convolution kernels is 16, 16, 64, 64, 256, 256, the activation function is the *relu* function, and the pooling layer is the maximum pooling and average pooling. The network structure is a stack of three consecutive convolutional layers and a maximum pooling, and finally an average pooling, regularization and fully connected layers are used for fault classification. The number of training rounds is 25 and the training time is 6 seconds, which meets the monitoring needs.

**5.4.1 Parameter tuning.** The selection of convolutional layers in this article is based on the structure and complexity of the data, and is continuously adjusted through experiments. For the situation where 12-dimensional features affect two faults, multi-layer convolution

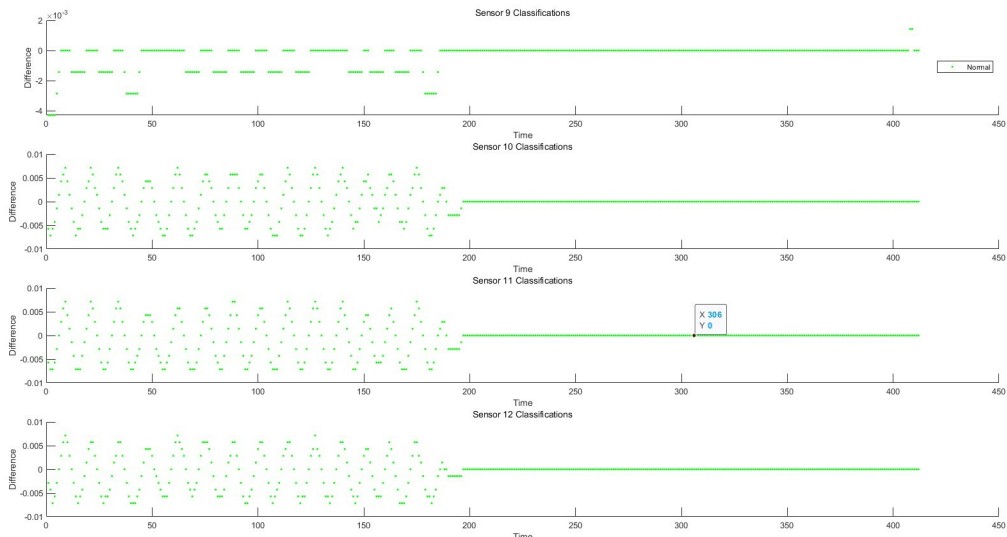

**Fig 11. Schematic diagram of 9–12 dimensional abnormal process monitoring.**

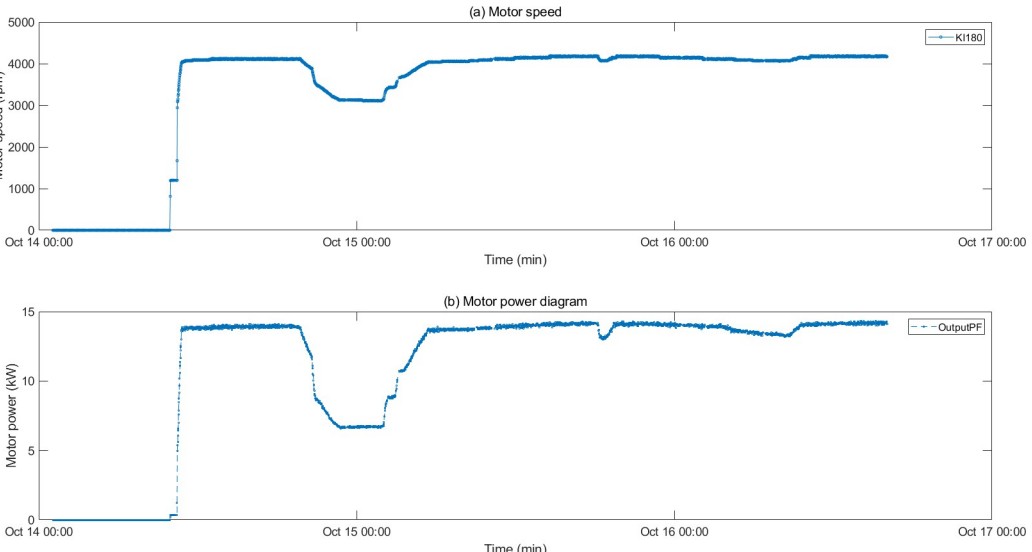

**Fig 12. Motor speed and power diagram.**

needs to be used to extract features at different levels. Through the combination of convolutional layers and pooling layers, high-level features can be effectively extracted. This paper uses two convolutions combined with one pooling to improve the generalization ability of the model. In addition, regularization techniques are combined to prevent overfitting, and triple stacking is performed. Small kernels can capture local features, and when multiple small kernels are stacked and used, global features can be captured at a deeper level. Large kernels can

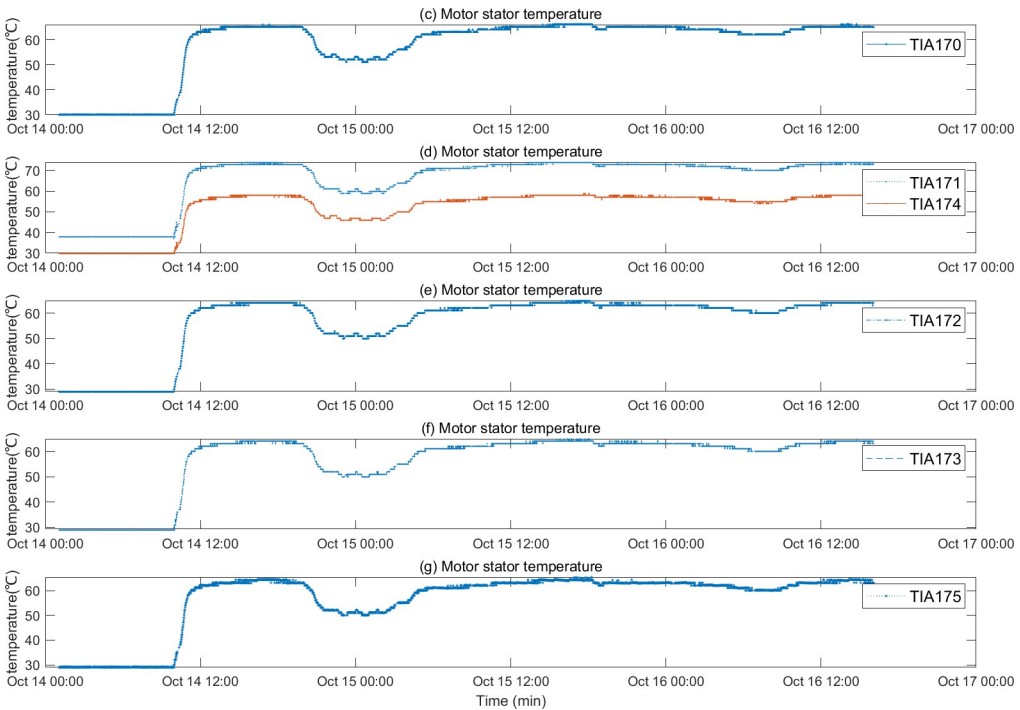

**Fig 13. Motor stator temperature diagram.**

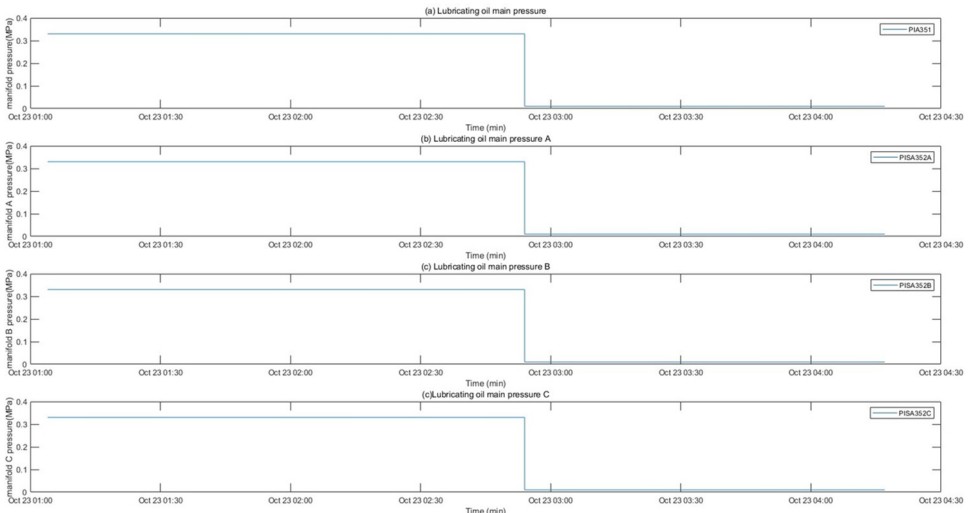

**Fig 14. Oil pressure change chart.**

directly capture a wider range of features, but require greater calculations. This paper prefers to use small kernels to deepen the layers to achieve the same effect. This method can effectively capture local features and has high computational efficiency. Many successful networks use small cores and gradually deepen the network, which is often an effective strategy. A lower initial learning rate of 0.0002 was chosen and the Adam optimizer was used to adapt to the learning rate. For the Dropout rate, 0.3 is a moderate Dropout rate that prevents overfitting while maintaining good model performance, and the number of filters is given an initial value space. Finally, grid search technology is used to optimize parameters.

## 5.5 Comparative analysis with other algorithms

The PCA-SPE-CNN, PCA-SPE-KNN, PCA-SPE-SVM, CNN classifiers have been validated with data and their performance has been evaluated. The accuracy, precision, recall, f1 score, and AUC-ROC and ROC curves were comprehensively evaluated by cross-validation with k-fold. The ROC curve is derived from the signal detection theory and represents the relationship between the true positive rate (TPR) and the false positive rate (FPR) of the classifier (TPR on the x-axis and FPR on the y-axis), and the validation results and evaluation results are described below.

Four additional important terms are also used in the calculation of various assessment parameters, namely true positive (TP), true negative (TN), false positive (FP), false negative (NP). True Positive Rate (TPR): The number of true positives (TP) divided by the number of samples that are true positive (TP + NP). False Positive Rate (FPR): The number of false positives (FP) divided by the number of samples that are truly negative (FP + TN). TP: These are positive datasets that are accurately labeled by classification methods. TN: These are negative datasets that are accurately labeled by classification methods. FP: These are negative datasets that have been incorrectly labeled as positive datasets by the classification method. NP: These refer to positive datasets that have been mislabeled as negative datasets by the classification method.

**5.5.1 Classification accuracy and performance analysis of PCA-SPE-CNN classifier.**
First, offline normal data is used for training, then a sliding window is used to perform fault monitoring on the monitoring data, and finally the monitored data is screened for fault data.

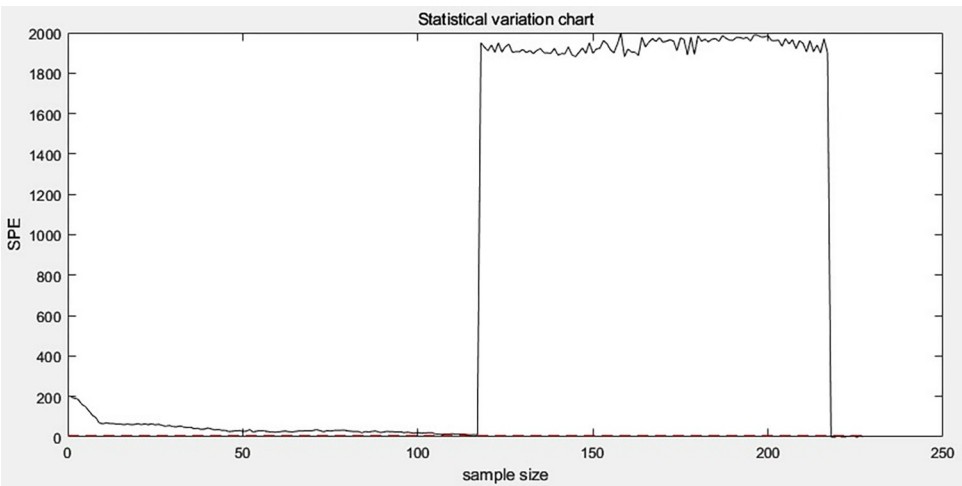

**Fig 15. Fault monitoring.**

According to Eqs (15) and (20), the SPE statistic $Q$ and the statistical limit $Q_\alpha$ of the monitoring data are calculated respectively, and the fault monitoring is carried out by comparing the magnitude of the $Q$ and $Q_\alpha$ values. The monitoring results are shown in Fig 15. Those that are higher than the limit value are fault data. The corresponding normal data filtering part is shown in Fig 16.

After training the PCA-SPE-CNN algorithm, the confusion matrix was constructed. The improved algorithm achieved 100% classification accuracy. The accuracy of the training set and verification set and the classification results of the model are shown in Fig 17. In addition, the model passed an evaluation based on the ROC curve. The ROC curve is derived from the theory of signal detection and represents the relationship between the true positive rate (TPR) and the false positive rate (FPR) of the classifier (TPR on the x-axis and FPR on the y-axis). This curve reflects the model performance under different thresholds, and the results are shown in Fig 18. The model underwent k-fold cross validation as shown in Fig 19.

In order to comprehensively evaluate the performance of the model, analysis was conducted on accuracy, precision, recall, F1 score, and AUC-ROC indicators. The results are shown in Table 1. And the Bootstrap method was used to estimate the performance metrics, so as to improve the robustness of the analysis.

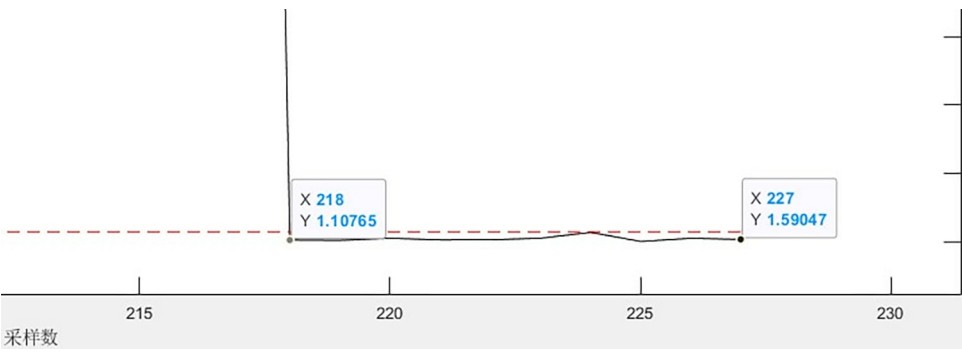

**Fig 16. Partial enlargement of normal data monitoring.**

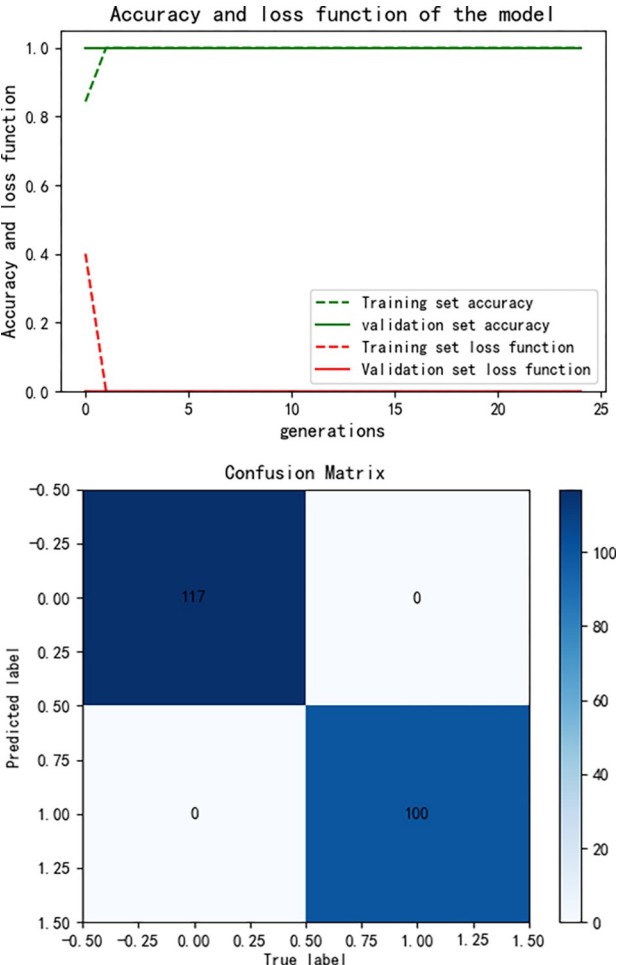

**Fig 17. PCA-SPE-CNN classification results.**

By resampling 1000 times, ensure that the model performs more robustly under different validation set samples. Finally, calculate the mean and 95% confidence interval of the performance indicators to measure the fluctuation of the model's performance under different sampling conditions. The calculated result is Mean AUC: 1.000,95% CI: [1.000,1.000]. Mean Accuracy: 1.00, 95% CI: [1.00, 1.000].Mean AUC: represents the average AUC value of the model among all Bootstrap resamples, with a value closer to 1 indicating better discriminative ability of the model. 95% CI for AUC: The 95% confidence interval of AUC, indicating that the AUC value of the model will fall within this interval with a 95% probability under different resampling conditions. If the interval is small, it indicates that the model performance is relatively stable; If the interval is large, it indicates that the model exhibits significant fluctuations under different data. Mean Accuracy: Refers to the average accuracy of the model's predictions across all resamples. 95% CI for Accuracy: The 95% confidence interval for accuracy, representing the fluctuation range of the model's accuracy under different resampling conditions.

**5.5.2 Classification accuracy and performance analysis of PCA-SPE-KNN, PCA-S-PE-SVM, CNN classifiers, and expert system diagnostic results.** Due to significant differences in the selected data, it is difficult to distinguish between known edge normal data and abnormal data. After training PCA-SPE-KNN, PCA-SPE-SVM, and CNN algorithms, a

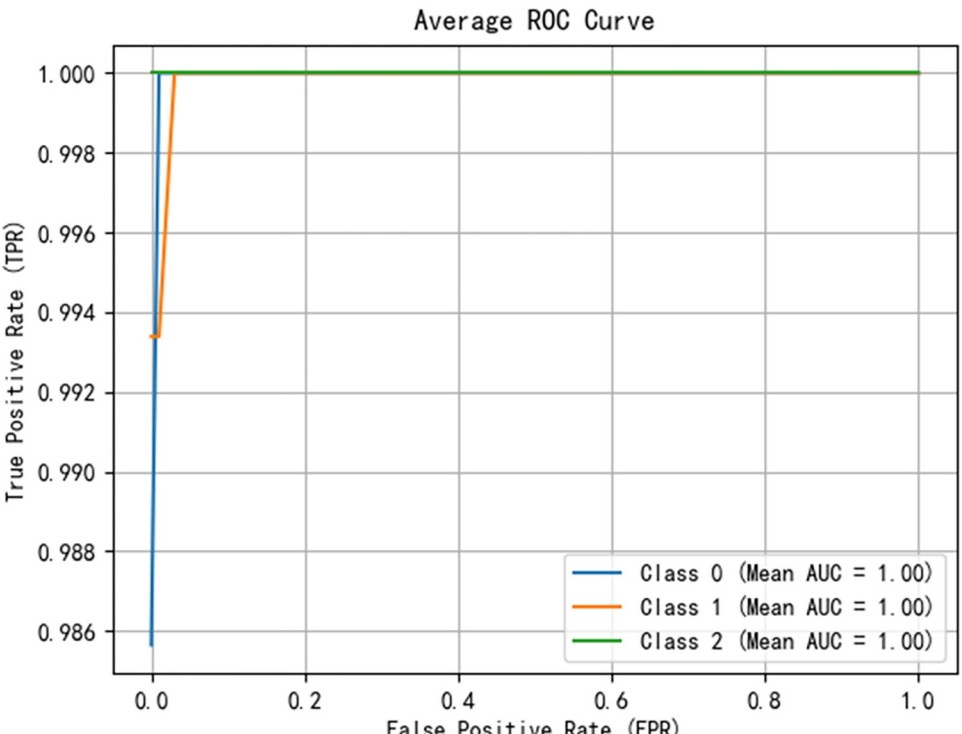

**Fig 18. ROC curve of training set validation set.**

confusion matrix was constructed, and complex models were compared based on accuracy, precision, recall, and F1 score. According to the confusion matrix, the classification performance is far inferior to the improved PCA-SPE-CNN model, and the accuracy, recall, and F1

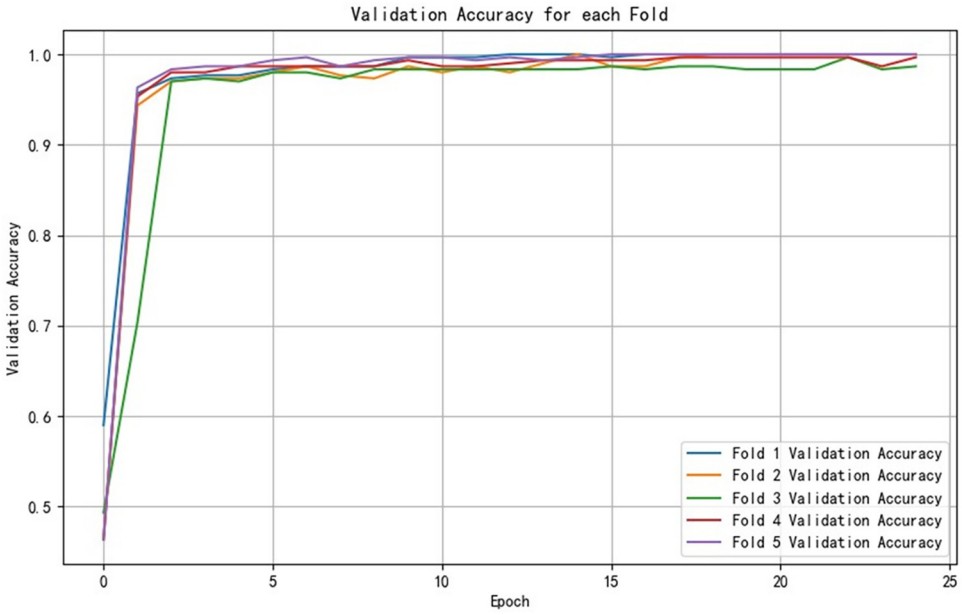

**Fig 19. Cross-validation results.**

**Table 1. Indicator evaluation.**

| index / Folded number | Validation accuracy | Accuracy | Precision | Recall | F1-Score | AUC-ROC |
|---|---|---|---|---|---|---|
| 1-fold | 1 | 1 | 1 | 1 | 1 | 1 |
| 2-fold | 1 | 1 | 1 | 1 | 1 | 1 |
| 3-fold | 0.997 | 0.987 | 0.987 | 0.987 | 0.987 | 1 |
| 4-fold | 0.997 | 0.997 | 0.997 | 0.997 | 0.997 | 1 |
| 5-fold | 1 | 1 | 1 | 1 | 1 | 1 |
| Mean | 0.998 | 0..997 | 0.997 | 0.997 | 0.997 | 1 |

score of the three compared models are not as good as the improved model. Figs 20–22 show their classification results, and Table 2 shows their metric evaluations. By comparing the above, the best solution can be achieved.

Expert system diagnosis results are shown in Fig 23 below.

## 6. Conclusions

This paper proposes and implements a safe and reliable sliding window fault-tolerant filter, as well as a residual generation and data mutation monitoring algorithm based on fault-tolerant filtering. The performance of the proposed PCA-SPE-CNN classifier is tested based on process data from an industrial centrifugal compressor. This classifier is also combined with an expert system based on knowledge graph and NLP inference engine to realize the causal traceability of the expert system. This method greatly improves the following problems:

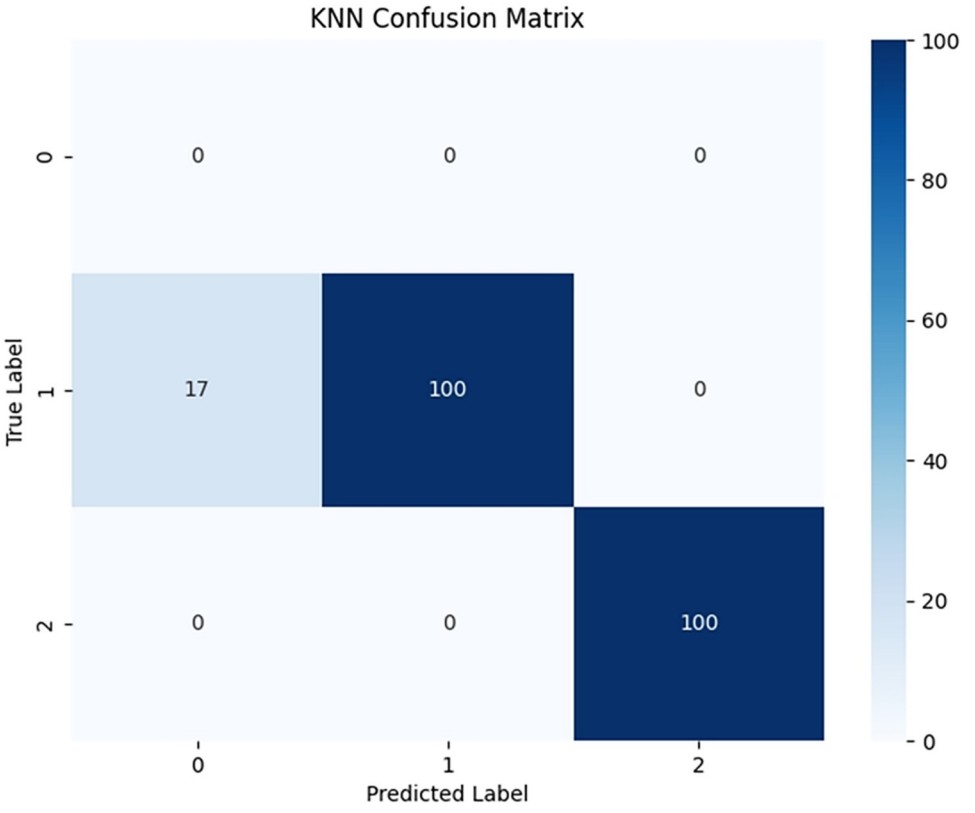

**Fig 20. PCA-SPE-KNN classification results.**

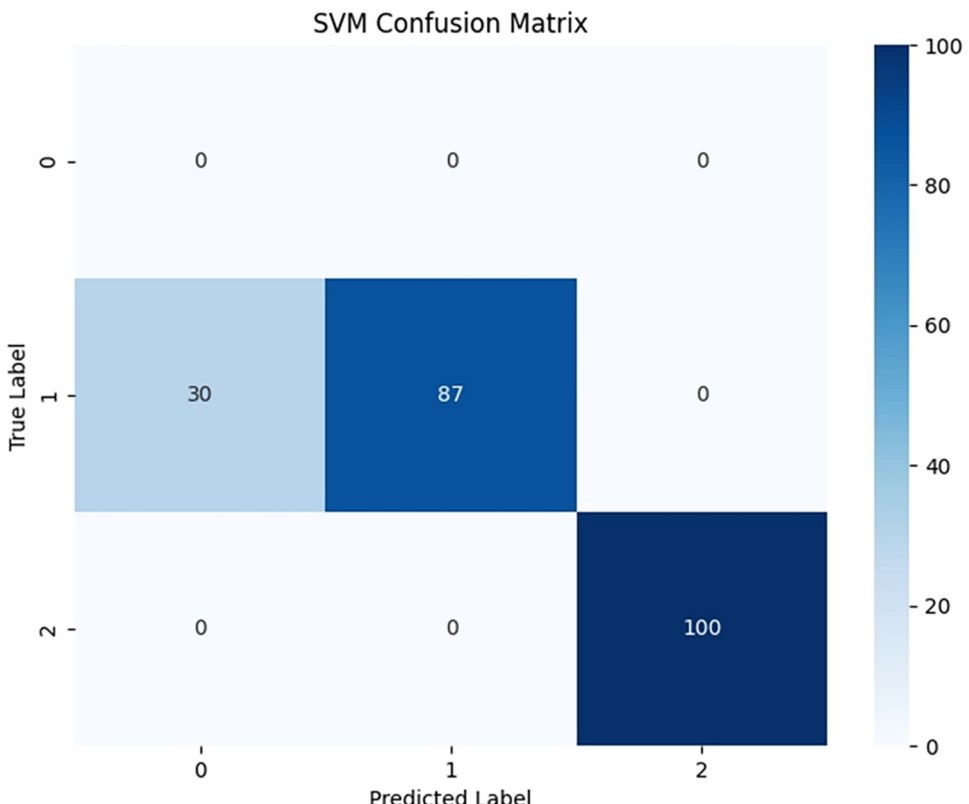

**Fig 21. PCA-SPE-SVM classification result.**

First, the above monitoring method can quickly realize early symptom detection, small-amplitude variance identification and fault isolation, effectively overcoming the existing DCS or FCS system instrumentation data monitoring commonly used based on a fixed threshold alarm method of false alarms and missed alarms, as well as difficult to realize the limitations of within-threshold variance monitoring and so on; The second is to improve the accuracy of fault classification; Third, it improves the problems of difficulty in knowledge base maintenance and the lack of flexibility of traditional rule-based fault diagnosis expert systems. The system is able to learn from actual operational data and adapt adaptively to changing circumstances. This system can effectively avoid the limitations of conventional forward reasoning and reverse reasoning that are difficult to apply to situations such as network correlation between causes and effects, and effectively overcome the multiple coupling interactions between detection devices, as well as between detection devices, the detected object, and the controlled process. The influence of the mutual relationship enables the traceability and location of instrument data and equipment faults.

**Table 2. PCA-SPE-KNN, PCA-SPE-SVM, CNN classifier metric evaluation.**

| name \ index | Accuracy | Precision | Recall | F1-Score |
|---|---|---|---|---|
| CNN | 0.955 | 1 | 0.953 | 0.973 |
| PCA-SPE-SVM | 0.862 | 1 | 0.862 | 0.921 |
| PCA-SPE-KNN | 0.922 | 1 | 0.922 | 0.958 |

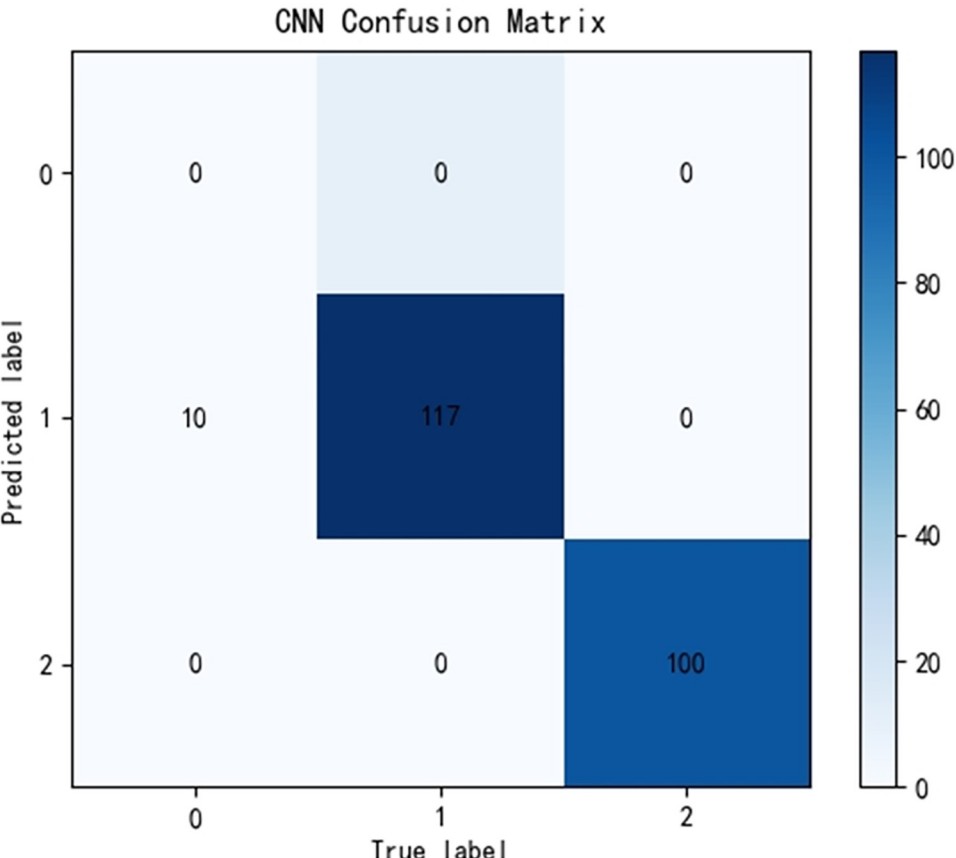

**Fig 22. CNN classification result.**

In future research, more advanced machine learning algorithms can be explored to improve the real-time and diversity of fault monitoring; Examine the applicability and effectiveness of this method in other industries such as manufacturing, and promote the widespread use of the technology.

```
['Driver does not start', 'Oil pressure too low']
model init finished ......

Hi! I am Dr.Yar, I am here to help you make your health better.
For that you'll have to answer a few questions about your conditions
Do you feel any of the following symptoms:

The most probable disease that you have is Oil_pressure_too_low

The most probable disease that you have is Driver_does_not_start

consult:Oil pressure too low reason
diagnostic robot: Oil pressure too low Cause of the failure: There is something wrong with the pump (the main oil pump does not start, the auxiliary oil pump does not s
consult:Driver does not start corrective measures
diagnostic robot: Driver does not start Corrective Actions: Oil pressure is too low: The pump is defective, the oil pipeline is leaking, the cooler, filter or strainer
consult:Oil pressure check examine
diagnostic robot: Oil pressure check Pre-startup inspection: The auxiliary oil pump must be automatically switched on when the lower oil pressure limit value is reached.
consult:main fuel tank  How to maintain
diagnostic robot: main fuel tank maintenance: Check the paint condition of the tank wall, check the bottom of the fuel tank, and clean it every time the oil is drained.
consult:
```

**Fig 23. Diagnostic results of driver not starting.**

## Supporting information

**S1 Data.**
(CSV)

**S2 Data.**
(CSV)

**S3 Data.**
(CSV)

**S4 Data.**
(CSV)

## Author Contributions

**Conceptualization:** Yuan Wang, Shaolin Hu.

**Data curation:** Yuan Wang.

**Formal analysis:** Shaolin Hu.

**Investigation:** Shaolin Hu.

**Methodology:** Yuan Wang, Shaolin Hu.

**Software:** Yuan Wang.

**Validation:** Yuan Wang.

**Visualization:** Yuan Wang.

**Writing – original draft:** Yuan Wang.

**Writing – review & editing:** Shaolin Hu.

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
