## [Decision Letter · Decision Letter 0]

27 Sep 2024

PONE-D-24-36704DESIGN AND IMPLEMENTATION OF COMPRESSOR DATA VARIATION SAFETY MONITORING AND INDUCEMENT TRACEABILITY EXPERT SYSTEMPLOS ONE

Dear Dr. Hu,

Thank you for submitting your manuscript to PLOS ONE. After careful consideration, we feel that it has merit but does not fully meet PLOS ONE’s publication criteria as it currently stands. Therefore, we invite you to submit a revised version of the manuscript that addresses the points raised during the review process.

We look forward to receiving your revised manuscript.

Kind regards,

Anurag Sinha, Ph.D

Academic Editor

PLOS ONE

Journal Requirements:

“This work was supported by the Natural Science Foundation of China (62373115); Nature and Science Foundation of Guangdong Province (2023A1515012341,2024A1515010870)”

Reviewers' comments:

Reviewer's Responses to Questions

**Comments to the Author**

1. Is the manuscript technically sound, and do the data support the conclusions?

Reviewer #1: Yes

Reviewer #2: Yes

Reviewer #3: No

2. Has the statistical analysis been performed appropriately and rigorously? 

Reviewer #1: Yes

Reviewer #2: Yes

Reviewer #3: No

3. Have the authors made all data underlying the findings in their manuscript fully available?

Reviewer #1: Yes

Reviewer #2: Yes

Reviewer #3: Yes

4. Is the manuscript presented in an intelligible fashion and written in standard English?

Reviewer #1: Yes

Reviewer #2: Yes

Reviewer #3: Yes

5. Review Comments to the Author

Reviewer #1: In this paper, the authors have conducted a study on the enhancement of compressor fault diagnosis by developing a knowledge-based expert system that integrates PCA-SPE-CNN, dynamic knowledge graphs, and natural language processing (NLP). While the research addresses an important problem in predictive maintenance for petrochemical equipment, a few suggestions are provided to enhance the clarity and depth of the methodology before recommending it for publication:

1. The authors need to revise the numerical enumeration of Sections in the manuscript. For example, the numbering of Section 2 is repeated twice.

2. While abbreviations like PCA-SPE and CNN are common in the field, providing a brief explanation of each in the abstract would help readers unfamiliar with these terms.

3. I would recommend combining PCA-SPE with other dimensionality reduction techniques or neural networks) which could possibly increase robustness in different operational conditions.

4. I would highly suggest the authors to add a Related Works section that provides information on the previous researches that were conducted in the same field and identify ways that this research has contributed to the previous ones. Perhaps, whether the research conducted was an idea obtained from previous work's future directions.

5. The authors should provide a more clear and high resolution image for Figure 14. The text is barely visible even when the image is largely scaled. Same goes for Figure 6.

6. Since the method is intended for real-time fault detection, discussing how it deals with noisy, incomplete, or anomalous data in real-world settings would be valuable.

7. I would suggest the authors to provide more details on the real-time updating mechanism of the dynamic knowledge graph. Specifically, explain how the system handles large, incremental updates and maintains accuracy without introducing delays or compromising system efficiency.

8. The authors should consider restructuring the sections so that there is a clear progression from data collection and processing to knowledge graph construction, and finally, to fault diagnosis and traceability.

Reviewer #2: The article titled “Design and implementation of compressor data variation safety monitoring and inducement traceability expert system”, proposes an approach for compressor data security monitoring and trigger source identification using a dynamic knowledge graph. The approach aims to overcome the limitations in current petrochemical safety monitoring technologies and conventional expert systems by addressing the challenge of distinguishing between instrument anomalies and equipment failures

As observed from the review of the manuscript, its topic is generally of interest to the readership. However, some comments and suggestions for improving the quality of the paper, are outlined below:

-It is strongly recommended that the authors either expand the introduction or add a new separate section reviewing related works in the field. This would help better substantiate the innovative aspects of the research and highlight how it addresses a specific gap, contributing to the existing body of knowledge. Attention should be paid to the fact that the references already included in this article are relatively limited and do not feature particularly research studies (i.e. from the last five years), constraining both the quality and the potential impact of the article.

-The methodology underlying the proposed approach is adequately defined, and its performance is sufficiently evaluated, yielding results that could enable other researchers to replicate certain aspects. Furthermore, the analysis of the methodology and the results of the proposed solution are well-supported by a sufficient number of effectively presented charts and tables.

-The authors are strongly encouraged to revise the structure of the manuscript.

In particular, careful attention has to be paid to the proper numbering of the sections. For instance, Section 2 is attributed both to “CONSTRUCTION OF A RESIDUAL GENERATION AND DATA VARIANCE MONITORING ALGORITHM BASED ON FAULT-TOLERANT FILTERING” and “CONSTRUCTION OF A DEEP LEARNING BASED SYSTEM FAULT CLASSIFIER”. Moreover, subsections should not include only a paragraph (i.e. subsection 4.4) or a Figure (i.e. subsection 4.6).

-Outlining potential directions for future research in the conclusions section, could enhance the impact of the research outcomes.

Structure and English Language: The paper is written in appropriate English language according to the standards of the Journal, however the authors are strongly recommended to conduct some proofreading and spell-checking.

Reviewer #3: -Abstract clearly states the challenge of “Fault diagnosis”, in oil and gas industry but information on models and performance metric is not clear.

-Data is not explained, descriptive stats about data is missing, shape of data, features in data, distribution of class etc.

-Page 25 – “the convolution kernel is 1*3, the number of convolution kernels is 16, 16, 64, 64, 256, 256, the activation function is the function, and the pooling layer is the maximum pooling and average pooling.” What is meant by the activation function is the function”?

-The introduction part is not explained well, no literature review.

-“hyperparameters not listed.

-Future study directions that concentrate on different tracks have not been added by the writers for the benefit of upcoming scholars who wish to further the academics in this sector.

6. PLOS authors have the option to publish the peer review history of their article (what does this mean?). If published, this will include your full peer review and any attached files.

Reviewer #1: **Yes: **Shraiyash Pandey

Reviewer #2: **Yes: **Prof. Konstantinos G. Arvanitis

Reviewer #3: No

---

## [Author Response · Author response to Decision Letter 0]

29 Oct 2024

Reviewer #1: 

Q1. The authors need to revise the numerical enumeration of Sections in the manuscript. For example, the numbering of Section 2 is repeated twice.

Response: Thank you for your detailed review and valuable feedback on the manuscript. I have carefully reviewed the manuscript and made corresponding modifications to address the issue of duplicate chapter numbers that you pointed out, such as Section 2 being duplicated twice..

Q2. While abbreviations like PCA-SPE and CNN are common in the field, providing a brief explanation of each in the abstract would help readers unfamiliar with these terms.

Response: Thank you for your careful review and valuable suggestions on my manuscript. I completely agree with your point about providing brief explanations of abbreviations in the abstract (such as PCA-SPE and CNN). Short explanations can indeed help readers who are not familiar with these terms better understand the content of the article. 

I have made the following modifications to the abstract: When PCA-SPE and CNN first appear in the abstract, their full names and brief explanations (principal component analysis squared prediction error) and (convolutional neural network) are provided.

Q3. I would recommend combining PCA-SPE with other dimensionality reduction techniques or neural networks) which could possibly increase robustness in different operational conditions.

Response: Thank you for your suggestion. After careful consideration and analysis, I have the following opinions on this suggestion:

1. Information redundancy and increased complexity: PCA is a linear dimensionality reduction method that, when combined with highly nonlinear techniques such as neural networks, may result in mismatched linear features, thereby increasing the complexity and tuning difficulty of the model. Especially in neural networks, the linear features generated by PCA may not effectively capture nonlinear features, which may instead introduce redundant information.

2. Computing resources and time overhead: The training process of neural networks itself requires a significant amount of computing resources and time. If combined with PCA-SPE, although PCA can reduce data dimensionality, it may further increase the training time and resource consumption of the entire model, especially when dealing with large-scale datasets.

3. Model interpretability issue: Although the results of PCA have some interpretability (each principal component represents certain linear features), the nonlinearity and black box nature of neural networks make interpretation difficult, which may reduce the interpretability of the model and affect the trust of experts and users in the results.

4. Risk of excessive reduction of information: Although PCA aims to preserve the most important features, continuing to use other dimensionality reduction methods after PCA may further reduce key information in the data. Even if each dimension reduction step removes redundant information, it may ultimately lead to excessive reduction of useful information, which has a negative impact on subsequent classification tasks.

Therefore, based on the above points, I have decided to maintain the original PCA-SPE method to avoid issues such as information redundancy, increased complexity, and decreased interpretability. However, your suggestion inspired me to explore in future research how to balance the combination of linear and nonlinear features while preserving model simplicity.

Thank you again for your valuable feedback. Looking forward to your further feedback!

Q4. I would highly suggest the authors to add a Related Works section that provides information on the previous researches that were conducted in the same field and identify ways that this research has contributed to the previous ones. Perhaps, whether the research conducted was an idea obtained from previous work's future directions.

Response: Thank you for your detailed review of my manuscript and valuable suggestions. I fully agree with your suggestion to add relevant works as an effective way to enhance the background information and academic contribution of the paper.

I have made the following modifications in the manuscript:

New references have been added to the preface, which provides a detailed introduction to previous work related to my research field, covering the latest research findings and key contributions. A summary was made of the core methods, results, and findings from these previous studies, and the differences and extensions of this study from these works were clearly identified.

Q5. The authors should provide a more clear and high resolution image for Figure 14. The text is barely visible even when the image is largely scaled. Same goes for Figure 6.

Response: Thank you for pointing out the quality issues with Figures 14 and Figures 6. I have replaced the higher resolution images for Figure 14 and Figure 6 to ensure that the text and image details are clearly visible.

Q6. Since the method is intended for real-time fault detection, discussing how it deals with noisy, incomplete, or anomalous data in real-world settings would be valuable.

Response: Thank you for the reviewer's suggestions. Our research mainly focuses on the actual operating data and corresponding fault cases of the compressor at Gaoling Station provided by China Petroleum West East Gas Pipeline Co., Ltd. I attach great importance to your suggestion on how to handle noise, incomplete or abnormal data in real environments using this method. In each stage of data processing, I applied fault-tolerant filtering algorithms for data processing, and used residual generation and data anomaly monitoring algorithms based on fault-tolerant filtering to monitor anomalies in each dimension of the data. This method demonstrates adaptability and robustness in practical complex environments. Thank you again for your valuable feedback. Looking forward to further feedback!

Q7. I would suggest the authors to provide more details on the real-time updating mechanism of the dynamic knowledge graph. Specifically, explain how the system handles large, incremental updates and maintains accuracy without introducing delays or compromising system efficiency. 

Response: Thank you for your suggestion. Regarding the process of using a graph database to store knowledge graphs and implementing iterative optimization, we have implemented the following steps:

1. Real time data reception: The system continuously receives instrument data and process data from the petrochemical DCS system to ensure the timeliness and integrity of the data.

2. Incremental data information processing: By combining incremental data, we can update the knowledge graph in real-time and achieve rapid iteration of knowledge.

3. PCA-SPE-CNN classifier: By constructing a PCA-SPE-CNN classifier, we classify faults in real-time monitored instrument data to ensure accurate identification of potential faults.

4. Combining knowledge graph with expert knowledge: We combine fault classification results with expert knowledge in the knowledge graph to promote knowledge updates and iterations, achieving a more efficient inference process.

The ultimate goal of this method is to continuously improve the intelligence level of the knowledge graph while ensuring system accuracy and efficiency. Looking forward to your further guidance.

Q8. The authors should consider restructuring the sections so that there is a clear progression from data collection and processing to knowledge graph construction, and finally, to fault diagnosis and traceability. 

Response: Thank you for your suggestion. We have made structural adjustments and look forward to your further guidance.

Reviewer #2: 

Q1. It is strongly recommended that the authors either expand the introduction or add a new separate section reviewing related works in the field. This would help better substantiate the innovative aspects of the research and highlight how it addresses a specific gap, contributing to the existing body of knowledge. Attention should be paid to the fact that the references already included in this article are relatively limited and do not feature particularly research studies (i.e. from the last five years), constraining both the quality and the potential impact of the article.

Response: Thank you for your detailed review of my manuscript and valuable suggestions. I fully agree with your suggestion to add relevant works as an effective way to enhance the background information and academic contribution of the paper.

I have made the following modifications in the manuscript:

New references have been added to the preface, which provides a detailed introduction to previous work related to my research field, covering the latest research findings and key contributions. A summary was made of the core methods, results, and findings from these previous studies, and the differences and extensions of this study from these works were clearly identified.

Q2. The authors are strongly encouraged to revise the structure of the manuscript.

In particular, careful attention has to be paid to the proper numbering of the sections. For instance, Section 2 is attributed both to “CONSTRUCTION OF A RESIDUAL GENERATION AND DATA VARIANCE MONITORING ALGORITHM BASED ON FAULT-TOLERANT FILTERING” and “CONSTRUCTION OF A DEEP LEARNING BASED SYSTEM FAULT CLASSIFIER”. Moreover, subsections should not include only a paragraph (i.e. subsection 4.4) or a Figure (i.e. subsection 4.6).

Response: Thank you for your suggestion. We have made structural adjustments and look forward to your further guidance.

Q3. Outlining potential directions for future research in the conclusions section, could enhance the impact of the research outcomes.

Response: Thank you for your in-depth review of our work. We have outlined the potential directions for future research in the conclusion section. Through the overview of these potential directions, we hope to provide inspiration for future research and lay the foundation for further practical applications.

Q4. Structure and English Language: The paper is written in appropriate English language according to the standards of the Journal, however the authors are strongly recommended to conduct some proofreading and spell-checking.

Response: Thank you for your meticulous review and feedback on our paper. We have conducted language proofreading.

Reviewer #3: 

Q1. Data is not explained, descriptive stats about data is missing, shape of data, features in data, distribution of class etc.

Response: Thank you for your feedback on our paper. We have added a section on "Data Cause Analysis" in the revised version to provide a detailed description of the source, shape, characteristics, and other relevant information of the data.

Q2. Page 25 – “the convolution kernel is 1*3, the number of convolution kernels is 16, 16, 64, 64, 256, 256, the activation function is the function, and the pooling layer is the maximum pooling and average pooling.” What is meant by the activation function is the function”? 

Response: Thank you for your valuable feedback. We will further clarify in the revised version that the description of "activation function" is a function.

Q3. The introduction part is not explained well, no literature review.

Response: Thank you for your detailed review of my manuscript and valuable suggestions. I fully agree with your suggestion to add relevant works as an effective way to enhance the background information and academic contribution of the paper.

I have made the following modifications in the manuscript:

New references have been added to the preface, which provides a detailed introduction to previous work related to my research field, covering the latest research findings and key contributions. A summary was made of the core methods, results, and findings from these previous studies, and the differences and extensions of this study from these works were clearly identified.

Q4. hyperparameters not listed.

Response: Thank you for your valuable feedback on our paper. We have added a section on "Parameter tuning" in the revised version to provide a detailed list of hyperparameters used, including learning rate, batch size, and regularization parameters.

Q5. Future study directions that concentrate on different tracks have not been added by the writers for the benefit of upcoming scholars who wish to further the academics in this sector.

Response: Thank you for your in-depth review of our work. We have outlined the potential directions for future research in the conclusion section. Through the overview of these potential directions, we hope to provide inspiration for future research and lay the foundation for further practical applications.

---

## [Decision Letter · Decision Letter 1]

19 Nov 2024

PONE-D-24-36704R1Design and Realization of Compressor Data Abnormality Safety Monitoring AND INDUCEMENT TRACEABILITY EXPERT SYSTEMPLOS ONE

Dear Dr. Hu,

Thank you for submitting your manuscript to PLOS ONE. After careful consideration, we feel that it has merit but does not fully meet PLOS ONE’s publication criteria as it currently stands. Therefore, we invite you to submit a revised version of the manuscript that addresses the points raised during the review process.

We look forward to receiving your revised manuscript.

Kind regards,

Anurag Sinha, Ph.D

Academic Editor

PLOS ONE

Journal Requirements:

Reviewers' comments:

Reviewer's Responses to Questions

**Comments to the Author**

1. If the authors have adequately addressed your comments raised in a previous round of review and you feel that this manuscript is now acceptable for publication, you may indicate that here to bypass the “Comments to the Author” section, enter your conflict of interest statement in the “Confidential to Editor” section, and submit your "Accept" recommendation.

Reviewer #1: (No Response)

Reviewer #4: (No Response)

2. Is the manuscript technically sound, and do the data support the conclusions?

Reviewer #1: Yes

Reviewer #4: Yes

3. Has the statistical analysis been performed appropriately and rigorously? 

Reviewer #1: Yes

Reviewer #4: Yes

4. Have the authors made all data underlying the findings in their manuscript fully available?

Reviewer #1: Yes

Reviewer #4: Yes

5. Is the manuscript presented in an intelligible fashion and written in standard English?

Reviewer #1: Yes

Reviewer #4: Yes

6. Review Comments to the Author

Reviewer #1: 1. The authors have done a well job in constructing the abstract as it is presented very effectively and summarizes the key contributions of the research

2. The authors have chosen 5 as the value for their default k-parameter in the sliding window method. Perhaps they should explain more as to why they have chosen 5?

3. The authors should explain more about the dataset that they have used for training and validation

5. To ease off the comprehension factor for readers, the authors should try to add more simple termed notes for the complex mathematical formulas that are presented in the manuscript

5. The authors should revise the citations and make sure that there is a consistent citation style throughout the manuscript

Reviewer #4: The research focuses on creating a security monitoring and root cause tracing method for compressor data anomalies which is a great initiative especially since security is fundamental in such cases. Below are comments in relation to a few suggestions:

- The authors are highly recommended to re-enumerate the sections properly. For example, first Section should begin with 1, and not 0. Additionally, it should be in a high priority since it was already stated in previous revision to re-enumerate properly.

- There is very minimal information provided in terms of overview even for major section headings such as Section 2.

- The authors are suggested to provide an overview for major headings such as Section 4 before straightly jumping to subsections, in this case, subsection 4.1.

- Why is subsection 4.6 left empty and there is no content presented within it?

7. PLOS authors have the option to publish the peer review history of their article (what does this mean?). If published, this will include your full peer review and any attached files.

Reviewer #1: No

Reviewer #4: No

---

## [Author Response · Author response to Decision Letter 1]

27 Nov 2024

Dear Editor:

 Thank you very much for giving us an opportunity to revise our manuscript, we appreciate you and the reviewers very much for their positive and constructive comments and suggestions on our manuscript entitled “Design and realization of compressor data abnormality safety monitoring and inducement traceability expert system”. 

We carefully studied the reviewers' comments and made revisions in a comparatively revised manner. We have tried our best to revise our manuscript according to the comments. Attached please find the revised version, which we would like to submit for your kind consideration.

We would like to express our great appreciation to you and reviewers for reviewing and evaluating our manuscript. Looking forward to hearing from you.

 Thank you and best regards.

Yours sincerely,

Yuan Wang

Corresponding author:

Name: Shaolin Hu

E-mail:hfkth@126.com

Dear reviewers:

Thank you for your excellent and insightful series of remarks. We have processed these comments you raised and made the modifications in the form of comparative revisions. We hope that the revision is acceptable and look forward to hearing from you soon.

Thank you and best regards.

Yours sincerely,

Yuan Wang

Corresponding author:

Name: Shaolin Hu

E-mail:hfkth@126.com

Reviewer #1: 

Q1. The authors have done a well job in constructing the abstract as it is presented very effectively and summarizes the key contributions of the research

Response: Thank you for the reviewer's recognition of the abstract section. We are pleased that you believe the abstract effectively summarizes the main contributions of the research. Your affirmation has inspired us, and we appreciate your support and recognition of our work.

Q2. The authors have chosen 5 as the value for their default k-parameter in the sliding window method. Perhaps they should explain more as to why they have chosen 5?

Response: Thank you for your careful review and valuable suggestions on my manuscript. In our study, the choice of k=5 was based on the following considerations: 

1. Choosing 5 is usually because it can provide a good balance in practical applications, effectively filtering out noise while preserving the overall trend of the data. 

2. The size of the sliding window directly affects the computational complexity. A smaller window width can usually avoid excessive computational overhead while ensuring filtering effectiveness.

3. We chose k=5 as a reasonable value obtained through experimental evaluation. After multiple experiments, k=5 provides a good balance between the temporal correlation of data and computational efficiency.

Q3. The authors should explain more about the dataset that they have used for training and validation.

Response: Thank you for your suggestion. We have provided a more detailed explanation of the application of the dataset and algorithm. The residual generation and data anomaly monitoring algorithm based on fault-tolerant filtering is mainly used to detect the stationarity of each dimensional feature. The monitoring data used is individual data for each dimension and does not affect each other. The construction of classifier algorithms mainly considers the impact of twelve dimensional features on second-class faults, and the monitoring data is a comprehensive consideration of twelve dimensional data to capture the correlation between different features and improve the classification accuracy of the model.

Thank you for your suggestions. These supplements and explanations will help to better understand the structure of the dataset and the design ideas of the algorithm.

Q4. To ease off the comprehension factor for readers, the authors should try to add more simple termed notes for the complex mathematical formulas that are presented in the manuscript.

Response: Thank you for your detailed review of my manuscript and valuable suggestions. More annotations have been added to complex formulas in the text.

Q5. The authors should revise the citations and make sure that there is a consistent citation style throughout the manuscript.

Response: Thank you for your detailed review of my manuscript and valuable suggestions. Modifications have been made.

Reviewer #4: 

Q1. The authors are highly recommended to re-enumerate the sections properly. For example, first Section should begin with 1, and not 0. Additionally, it should be in a high priority since it was already stated in previous revision to re-enumerate properly..

Response: Thank you for your detailed review of my manuscript and valuable suggestions. Modifications have been made.

Q2. There is very minimal information provided in terms of overview even for major section headings such as Section 2.

Response: Thank you for your suggestion. We have made revisions and look forward to your further guidance.

Q3. The authors are suggested to provide an overview for major headings such as Section 4 before straightly jumping to subsections, in this case, subsection 4.1.

Response: Thank you for your suggestion. We have made revisions.

Q4. Why is subsection 4.6 left empty and there is no content presented within it?

Response: Thank you for your feedback on our paper. We have made revisions.

---

## [Editor Report · Decision Letter 2]

4 Dec 2024

Design and Realization of Compressor Data Abnormality Safety Monitoring And Inducement Traceability Expert System

PONE-D-24-36704R2

Dear Dr. Hu,

We’re pleased to inform you that your manuscript has been judged scientifically suitable for publication and will be formally accepted for publication once it meets all outstanding technical requirements.

Kind regards,

Anurag Sinha, Ph.D

Academic Editor

PLOS ONE

Additional Editor Comments (optional):

Author have incorporated all concerns raised by reviewers and thus, paper can be accepted in current form no further revision is required
---

## [Editor Report · Acceptance letter]

5 Dec 2024

PONE-D-24-36704R2 

PLOS ONE

Dear Dr. Hu, 

I'm pleased to inform you that your manuscript has been deemed suitable for publication in PLOS ONE. Congratulations! Your manuscript is now being handed over to our production team.

Kind regards, 

on behalf of

Mr. Anurag Sinha 

Academic Editor

PLOS ONE